# Mechanical and Electrical Simulations of the Tulip Contact System

**Sebastian Łapczyński [1], Michał Szulborski [1],*, Karol Gołota [2], Łukasz Kolimas [1] and Łukasz Kozarek [3]**

[1] Institute of Electrical Power Engineering, Warsaw University of Technology, 00-662 Warsaw, Poland; seb.lapczynski@gmail.com (S.Ł.); lukasz.kolimas@ien.pw.edu.pl (Ł.K.)

[2] KGMG Sp. Z O. O., 22-100 Warsaw, Poland; karolgolota@gmail.com

[3] ILF Consulting Engineers Polska Sp. Z.O.O., 02-823 Warsaw, Poland; lukaszkozarek@gmail.com

* Correspondence: mm.szulborski@gmail.com; Tel.: +48-662-119-014

**Abstract:** The purpose of this work is to discuss the tulip contact behavior during mechanical and electrical simulations in a Finite Element Method (FEM) environment using ANSYS and COMSOL software. During the simulations, the full contact movement was analyzed. During the contact movement, the individual behavior of the contact components was taken into consideration. the motion simulation was carried out at different velocities and forces acting on the contact. the obtained results were compared to each other and discussed. Relatively, the angles of the contact surfaces to each other were also changed, which meant that we could conduct a more in-depth analysis. the other approach of simulation research was a field analysis of physical phenomena occurring in the tulip contact. This analysis was performed in COMSOL Multiphysics. Parametric analysis allowed an observation of the electric field in the tulip contact at different contact distances with respect to each other. This work is important in terms of the cost effectiveness for design procedures concerning tulip contacts and fault avoidance, which both result from mechanical and electrical conditions throughout contact exploitation and optimization of the working conditions for the tulip contact.

**Keywords:** tulip contact; electrical contacts; FEM; simulation; mechanical analysis; electrical parameters

## 1. Introduction

Digital calculation methods such as the Finite Element Method (FEM) allow many calculations to be performed in parallel. This can significantly accelerate the whole design process of the chosen components. Digital techniques are also used in designing structures of electrical apparatuses. Using CAD software, it is possible to design devices from a single element to whole complex assemblies [1,2]. Simulations of such structures are also possible in a further stage, as part of a project using specialized software [3,4]. These can be employed in order to simulate the movement of elements relative to each other, exclude collisions, and end by observing the drafted system in conditions simulating a natural work environment [5,6].

The contacts of electrical apparatuses belong to the most loaded elements of current circuits [7,8]. Therefore, they should be designed, constructed, and operated in such a way that the permissible limitations of their technical parameters, resulting from relevant regulations and standards, are not exceeded [9]. the parameters of the electrical apparatus connecting process strictly depend on the parameters and kinematic, dynamic, and structural properties of the mechanism that is responsible for the movement of the electrical contact. the tasks of this mechanism include reliable movement of

the set of movable contacts in an intended manner and maintaining contact pressure in their contact state [10]. the mechanical part of the switching contacts is required to reliably maintain the contact state, regardless of the rated parameters of the apparatus. the temperature of the entire structure must be maintained at a level that allows stable working conditions, which are also specified in the relevant standards. the mechanism should be resistant to the phenomenon of contact clogging when connecting, operating, and exposed to fault currents, as well as when conducting analysis under short circuit conditions [11]. This is directly related to the working temperature of the contacts and the phenomenon of contact bounce. It is also important to minimize contact expenditure while switching on currents. a number of forces act on the contacts during the bonding process [12,13]. These can be divided according to the source of origin. If there were no difference in electrical potential between the contacts, the current would not flow while the contacts touched, and therefore, the associated forces would not appear.

The combination of two types of analyzes—motion analysis and analysis of the electric field distribution—gives the image closest to the real environment of the system in which tulip contact is usually employed. This allows the entire contact to be designed in the most optimal way. the optimization applies to both the design time and the costs and quantity of the prototype construction. Therefore, our team executed a number of simulations in order to prove that proper simulations can aid the design process of electrical contacts, fault detection, and optimization [14].

This work is novel as it focuses on an analysis of the operation of the fixed contact. Nevertheless, movable contact analysis was also taken into consideration and executed whilst discussing the simulation results. the following points highlight the impact of this work:

- the ability to observe the behavior, operation, forces, and energy of all lamellas of the stationary (fixed) contact;
- Observation of the influence of the eccentricity of the movable contact mounting on the possibility of switching on;
- Mechanical analysis of the contact as for a real physical system;
- Determination of electrical parameter values during the switching operation;
- Contact development in other extinguishing environments (future work on the procured model).

## 2. State of the Art

The rapidly growing demand for electrical devices (the main elements are the contact system and the extinguishing chamber) consisting of high-quality electrical components has led to a new approach to designing switches. the main goal of designing and constructing electrical devices is to create solutions that can significantly improve their electrical and mechanical parameters, size, and short-circuit resistance [15]. Modern electrical apparatus is built in accordance with applicable standards (e.g., European standards). These requirements also apply to high-voltage circuit breakers ($SF_6$ insulated). Despite the increasingly smaller sizes of the devices mentioned, they are adapted to the effective breaking of short-circuit currents and protection of other elements of the power system. Complex mechanical phenomena occur in the circuit-breaker during the closing and opening process. This affects the structural components of the switch (copper, tungsten, steel, and aluminum) [16].

Competing authors have mostly concentrated on only simulating the contact system from a single, uniform material [17–19]. In fact, with a small number of cuts, the stationary contact is massive and no overall mechanical analysis may be required [20]. Therefore, the analysis of phenomena can be reduced to only a single lamella [21]. Nevertheless, the example of a multi-lamella contact system, consisting of a large number of elements, requires a different approach. the importance of mutual displacements of lamellas that mutually interact was observed. This results in a change in the total energy, torque, and mechanical stresses that must be taken into account. It is worth noting that high-current tulip contact systems are also commonly used in the vacuum technique. It is true that in the vacuum technique, multi-lamella contact systems are always closed without voltage (the vacuum circuit breaker

is inserted into the distribution box on the grid). Nevertheless, the mechanical stress and the associated risks, including damage, failure, and destruction, are very similar to the systems in the extinguishing environment with $SF_6$ gas.

The FEM model presented in this paper is applicable to contact systems in various environments and geometric configurations. the results enable the proper construction of complex contact systems for high-voltage conditions (heavy duty). the presented solution enables the validation of advanced analytical calculations, and the implementation of different, often complicated, geometries in relation to the simplified models of contacts. the model enables the determination of values for scientific and research calculations.

## 3. Contacts in Electrical Apparatuses and Current Circuits

### 3.1. Non-Connecting and Connecting Contacts

Current circuits and contact systems allow the conductive operating of currents, and contact systems of electrical connectors are used to make connections in electrical circuits. a contact is a part of the current circuit in which the current flow is made possible by the conjunction of two conductors. Due to their function in the current circuit, a distinction can be made between connecting and non-connecting contacts [22]. Connecting contacts consist of movable and fixed elements, while non-connecting contacts are only made of fixed elements. an example of connecting and non-connecting contacts is shown in Figure 1 below.

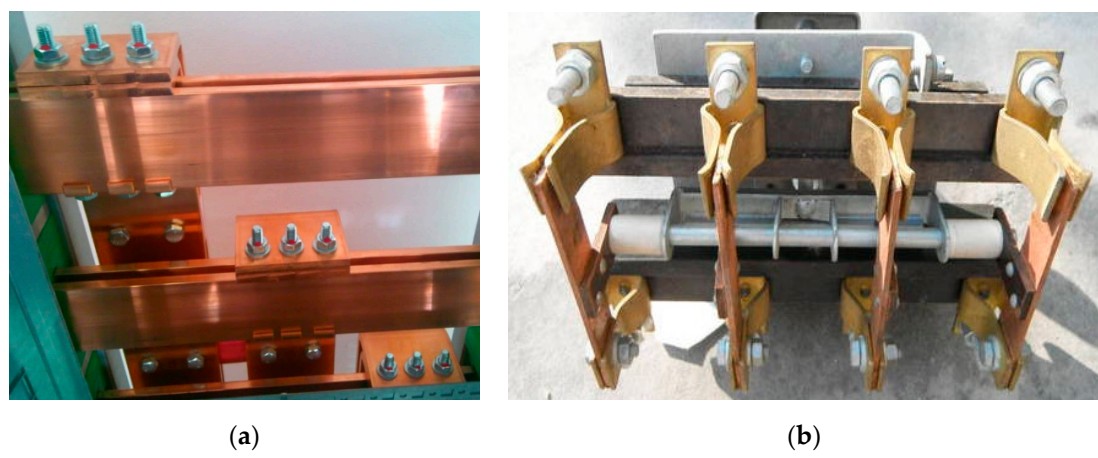

(**a**)          (**b**)

**Figure 1.** The view of exemplary contacts from families: (**a**) Non-connecting contacts (own study) and (**b**) connecting contacts [23].

Non-connecting contacts are fixed, and their elements cannot move with respect to each other. Figure 1a shows typical non-connecting contacts used in low-voltage switchgears. the movable contacts are used in contact switches and enable switching on and off of the current in a given circuit (such a contact is shown in Figure 1b). These can be divided into normally open (closed in the forced position of the movable contact) and normally closed (open in this position) contacts. Due to the shape of the contact surfaces, the contacts are divided into point, line, and surface contacts. a point contact is a contact in which contact takes place on a surface with a very small radius (through which current flows). the surface contact is characterized by a relatively large apparent (nominal) contact surface, while the actual contact surface of such a contact is a small percent of the apparent surface.

The contacts of electrical apparatuses belong to the most loaded elements of current circuits. Therefore, they should be designed, constructed, and operated in such a way that the permissible limits of their technical parameters resulting from relevant provisions and standards are not exceeded.

*3.2. Tulip Contacts*

Tulip contacts are an example from the group of connecting contacts. These contacts are also classified as coronary contacts and are often used in medium-voltage switchgears [23].

## 4. Construction of a Tulip Contact Used in Finite Element Method Analysis

The tulip contact which was used was patented under the number US 2012/0129374 A1/B2. the view of the tulip contact on which the three-dimensional modeling was based is shown in Figures 2 and 3 below.

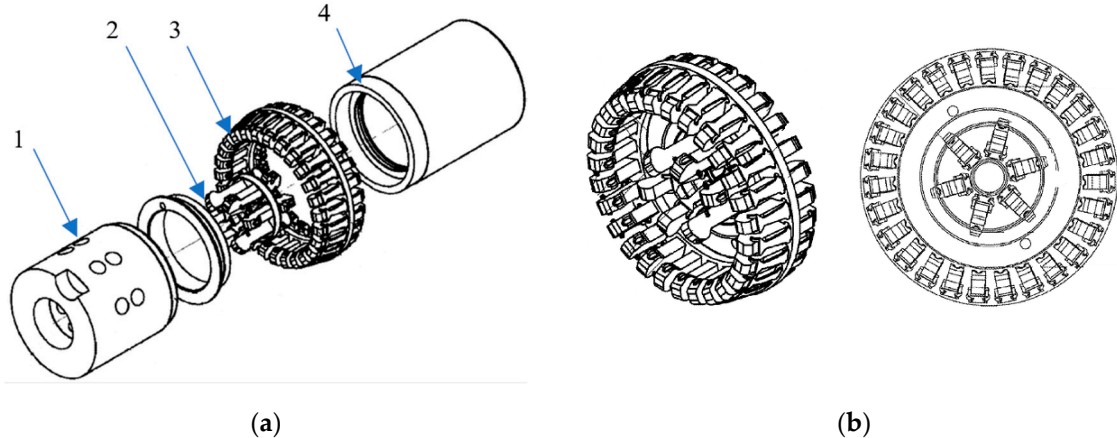

(**a**)　　　　　　　　　　　　　　　　　　　　　　　　　　　(**b**)

**Figure 2.** Structural view of the tulip contact: (**a**) 1—movable part of the contact, 2—inner crown lamellas, 3—outer crown lamellas, and 4—lamella seating, and (**b**) detailed view of tulip contact lamellas [24].

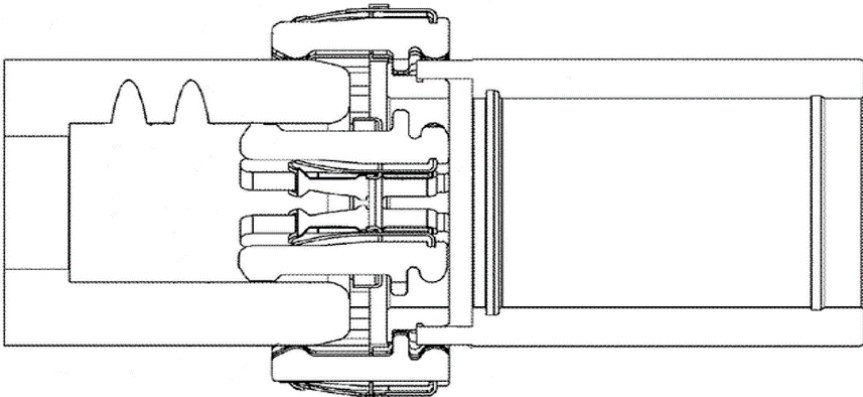

**Figure 3.** Longitudinal section of a closed tulip contact [24].

3D modeling was carried out in SolidWorks software. This software provides the ability to perfectly map the elements that were selected for simulation. Figure 4 was derived based on the materials proposed by the manufacturers of electrical apparatus and engineering experience. Three-dimensional models were mapped, which were used for further simulations in the ANSYS FEM software.

The movable contact element during the simulation of contact motion in the ANSYS software was assigned the physical and mechanical properties of a copper alloy. the same properties were also assigned to all lamellas. However, both fastening crowns (large crowns and small crowns) were defined as steel. Each electrical contact must be designed to meet electrical conditions in the open position and electro-mechanical conditions while closing and opening. All of the material data used in the simulation are described in Table 1.

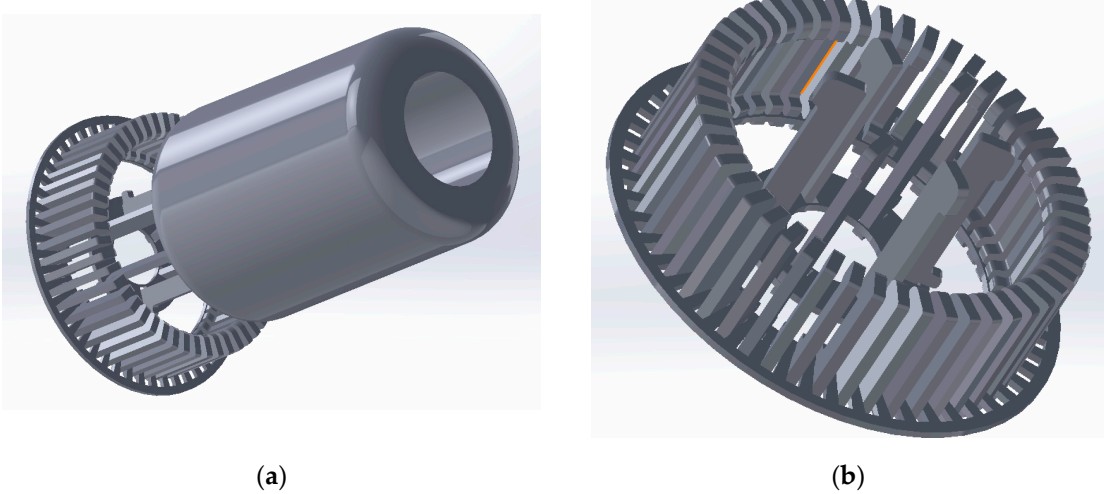

(**a**)           (**b**)

**Figure 4.** 3D model of the tulip contact: (**a**) View of a designed tulip contact model and (**b**) view of the lamellas in the tulip contact model.

**Table 1.** Table of the material data used in the procured model.

| Engineering Data | Copper | Steel | Unit |
|---|---|---|---|
| Density | 8300 | 7850 | $(kg/m^3)$ |
| Young's Modulus | $1.10 \times 10^{11}$ | $2.00 \times 10^{11}$ | (Pa) |
| Poisson's Ratio | 0.34 | 0.3 | |
| Bulk Modulus | $1.15 \times 10^{11}$ | $1.67 \times 10^{11}$ | (Pa) |
| Shear Modulus | $4.10 \times 10^{10}$ | $7.69 \times 10^{11}$ | (Pa) |

## 5. Physical Properties of Tulip Contact Systems

### 5.1. Insulation Strength of Contact Systems

The insulation strength mainly depends on the distribution of the electric field and the medium between the contacts. the layout of the contact and the materials used directly affect the electric field distribution. the electric medium should be air, gases, gas mixtures, and various types of oils.

Air is the most common gas dielectric used in practical high-voltage insulation systems. Its dielectric strength depends on many parameters. the most important are the construction of the insulation system and the electric field distribution, depending on the spark gap, type and course of voltage over time, air density, temperature, and humidity.

Sulfur hexafluoride has a number of properties that make it great as dielectric fluid. It is a very good insulating medium. It has a 2.5 times higher puncture strength than air, and at a pressure above three bar, its puncture strength is higher than that of mineral oil. Due to the high electron affinity of fluorine, the use of $SF_6$ increases the initial voltage of partial discharges. It is a very good extinguishing medium. For $SF_6$, the high electron affinity of fluorine combined with the abundance of fluorine on each discharge path ensures a strong interaction with high-energy electrons. This makes $SF_6$ almost 100 times more efficient at extinguishing than air. $SF_6$ is an inert and non-flammable gas. Only temperatures of 500 °C or electrical discharge can initiate dissociation and possible reactions. Fortunately, in the absence of other compounds, the dissociation products of $SF_6$ naturally combine with each other to regenerate $SF_6$. $SF_6$ is characterized by its unique ability to self-regenerate among dielectric fluids. Sulfur hexafluoride is an excellent heat carrier. the thermal conductivity coefficient of $SF_6$ is higher than that of air. Moreover, it grows with an increasing pressure. However, for sufficiently high pressures or flows, the $SF_6$ heat transfer coefficient is better than that of mineral oil. $SF_6$ is a non-toxic gas. This increases its attractiveness from the point of view of health and safety.

In practice, there is usually a channel mechanism for the development of discharge, because the electrode spacing usually ranges from 1 to 2 cm, which, at the atmospheric pressure and temperatures encountered in practice, is the border separating the two discharge mechanisms: Townsend and channel mechanisms.

Practical systems should usually be treated as blade systems, because the ratio of the distance to the dimensions of the electrodes is generally large and even avoiding sharp edges does not eliminate the unevenness of the electric field distribution. the dependence of the blade system strength on the electrode distance is shown in Figure 5 below.

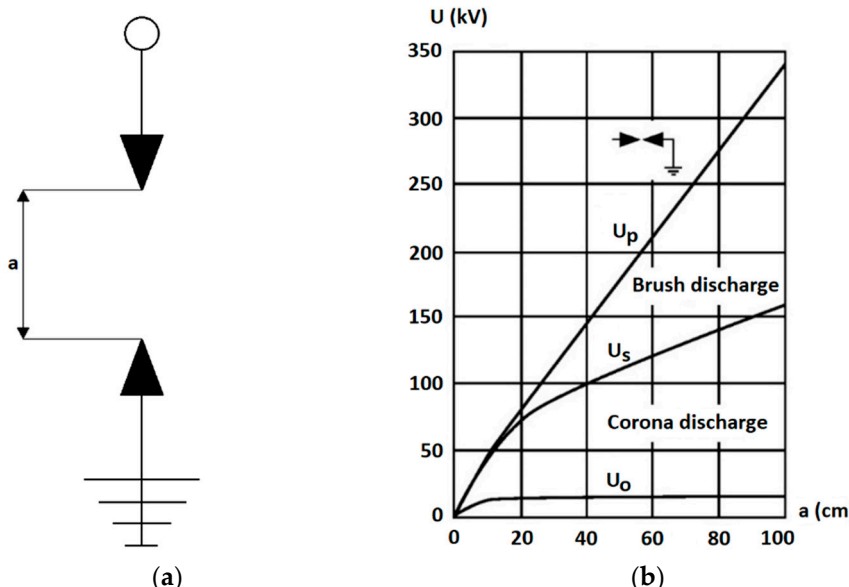

(a)                                                            (b)

**Figure 5.** View of a blade-type insulation system: (**a**) Blade with insulation distance a and (**b**) dependence of voltage $U_o$, $U_s$, and $U_p$ on the distance between the electrodes for the blade system, where $U_o$—initial voltage, $U_s$—shearing voltage, and $U_p$—jump voltage.

During the flow of electric current, electromagnetic forces occur in the contact system. These forces largely depend on the ratio between the radius of the external contact-R and the radius of the narrowing r, as well as on the position of the tangent point in relation to the entire system. As the number of contact points increases, the electrodynamic forces acting on the contacts decreases. the increasing distance between the contact points also lowers the force value. the solution used in the tulip contact is therefore a perfect example of dividing the contact points and providing a gap between them to counteract the electrodynamic force. When no current flows through the contact system, forces are still present in the process of closing and opening the movable contact. Along with the driving force closing and opening the movable contact, resistance forces occur. They result from movement of the movable contact surface against the stationary contact surface. the value of the resistance force depends on the contact material, the shape of the contacting elements, and the number of lamellas. an electrical apparatus installed in the power grid should cause the least losses, i.e., its contact resistance should be as low as possible. This requires the closed contact force to be maintained at a sufficiently high value. On the other hand, it is expected that the value of the resistance force when closing the contacts is low and as close as possible to that while opening the contact layout.

*5.2. Continuous, Variable and Short-Circuit Current Carrying Capacity*

In the process of designing and testing current apparatus, the only aim is to determine the current carrying capacity. the current carrying capacity in the process of electrical apparatus construction is a set value and also a design assumption. However, during the testing of existing switching devices, the heat dissipation capability at rated and short-circuit currents must be checked. the load

capacity of the cables largely depends on the intensity of heat dissipation from the heated body. Due to the conditions of heat dissipation from apparatus currents and switchgears, the following main calculation cases can be distinguished:

- Unprotected current circuits placed in air or an $SF_6$ environment, where heat is mainly released into the environment through radiation and lifting;
- Homogeneous current paths, surrounded by a layer of solid insulation, where all forms of heat transfer are significant;
- Heterogeneous current circuits, in which, in a steady state, an important role in heat transfer is carried out by axial heat flow.

In electric apparatuses and current circuits, the elements in which the resistance changes show the greatest heat gain. Such a change is usually observed at the contact points or narrowing of the current circuit. Figure 6 shows the heating of the current circuit from the point of electrical contact.

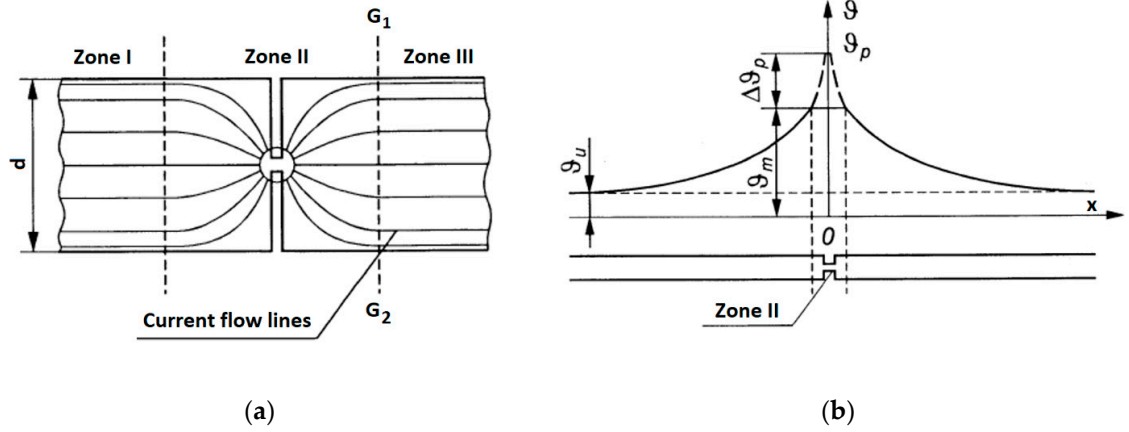

|                  |                  |
|:----------------:|:----------------:|
|       (a)        |       (b)        |

**Figure 6.** Heating of the current circuit with a single point contact: (**a**) Contact model and (**b**) temperature distribution along the current path, where ŭ—maximum temperature present at the contact point.

The heating constant of a given current circuit or contact system should also be determined. Several calculation methods can be employed for determining the characteristics. Below are some of the basic mathematical relationships used to determine the characteristics and the temperature waveforms as a function of time. the current circuit heating characteristic is shown in Figure 7.

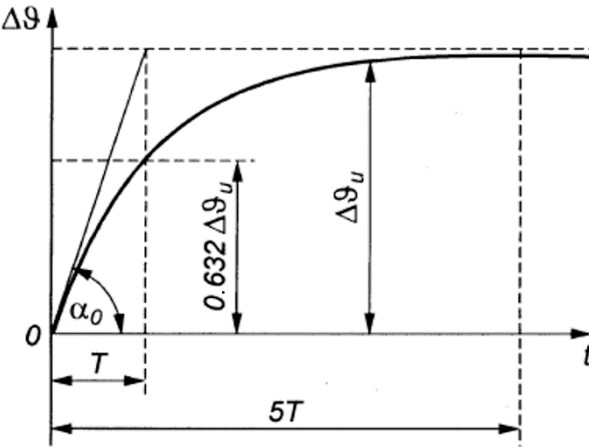

**Figure 7.** Typical characteristic of current circuit heating.

Current circuits can be subjected to continuous, occasional, and intermittent current loads. In all cases, the heating of the current circuits begins with the value of the ambient temperature.

the temporary and intermittent loads last for too short a period of time to determine the temperature rise. the interrupted load is a variable load with repeated periods and interruptions in a current flow. During load breaks, the conductor does not cool down to the ambient temperature because the no-current periods are too short compared to the value of time constant T.

## 6. Motion Simulations of the Tulip Contact System

Simulation studies were focused on by examining the dynamics of movement and physical phenomena in the tulip contact system. the analysis of the contact operation in the form of simulations was carried out in the ANSYS software. During the simulation, forces and times prevailing in existing tulip contacts were used. This resulted in obtaining values that are as close as possible to engineering experimental data. Several simulations were carried out relative to reference values. Both the forces arising from the release from constraints and the closing velocity of the contact were changed.

### 6.1. Environment for Simulation Research

ANSYS software was the tool used to perform motion analysis. the program used the Explicit Dynamics module, which is used to study the movement of fast-changing and short-lived events. the module analyzes phenomena and forces with non-linear characteristics. the internal structure of materials is analyzed during the interactions between elements and surface effects of motion dynamics. the tool used to perform the electrical analysis was COMSOL Multiphysics version. One of the calculation modules employed was the AC/DC module, which is used to simulate the electric, magnetic, and electromagnetic fields for static and low-frequency applications.

### 6.2. Discretization of the Procured Model

Extracting elements in area V, in which the solution is searched for, is a very important stage in creating the FEM calculation model. the method of discretization depends on the geometry of the area, physical properties, and certain general premises regarding the results of the solution, as well as the expected efficiency of calculations.

The method of discretizing the area determines the number of unknowns and the size and shape of the elements, and this affects the accuracy of the task solution. In order to obtain the required accuracy of the solution sought, the elements used should be small enough that the approximated functions inside them can be rough by means of polynomials. However, reducing the elements leads to an increase in the number of values of the nodal value function sought, and this simultaneously results in a longer calculation time. Most often, an uneven division into elements is used. While predicting where the function changes rapidly allows the element mesh to be compacted, where the function changes slowly, the element mesh should be diluted.

The accuracy of the solution primarily depends on the accuracy of the approximation of physical quantities inside the element using interpolation functions, hereinafter referred to as shape functions. With the correct mapping of the physical quantities of the element, reducing the areas of the elements (increasing their number) causes the nodal values of the sought function, which are an approximate solution of the task. the mesh process of the procured model is shown in Figure 8.

### 6.3. Analysis of the Tulip Contact Motion Dynamics—Variant I

The figures below show the mechanical stresses resulting from the motion analysis performed. By means of color spots, the force values are graphically represented, which can be read from the scale on the left side of each drawing. Graphical presentation of the results significantly speeds up the analysis of the results and facilitates the work. Figure 9 shows the entire tested tulip contact from two calculation views. Each of them unambiguously showed the highest loads in this type of tested structure. Thee mechanical loads appear in small movable elements, i.e., small and large lamellas and crown fixing of these lamellas. It should be remembered that in commercially produced contacts, each lamella is pressed towards the contact axis by springs surrounding the entire tulip or through direct spring pads

that press against a single frame. In the analyzed contact, such elements were replaced by pressing forces applied in appropriate places, allowing the system to be released from such constraints.

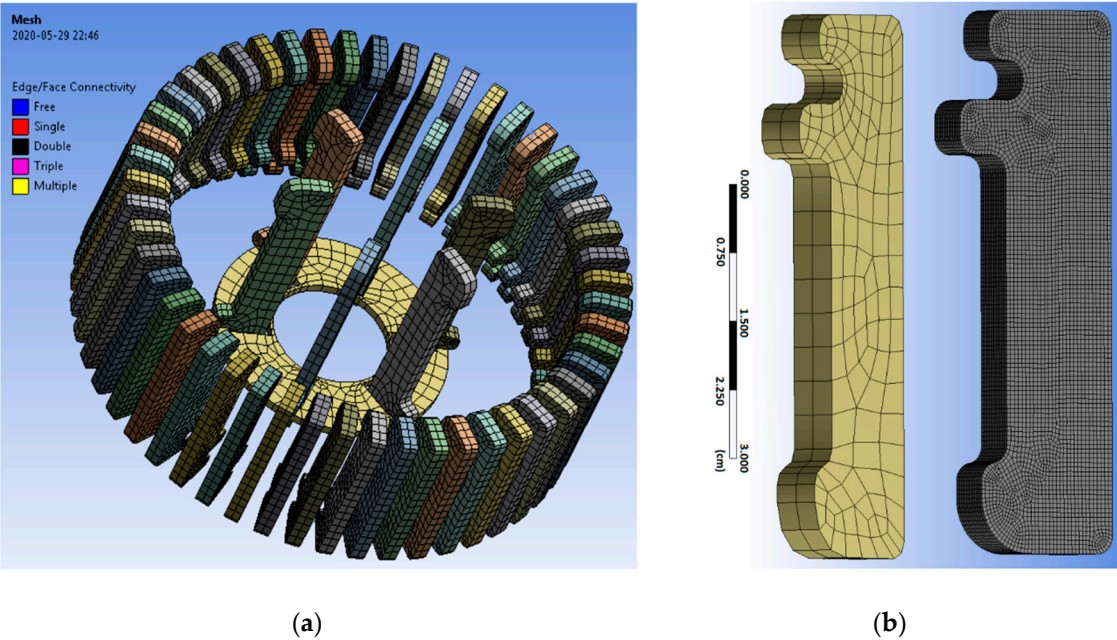

(**a**)  (**b**)

**Figure 8.** Meshing procedure of tulip contact system elements: (**a**) Tulip contact and (**b**) lamella with different rates of mesh during the discretization process.

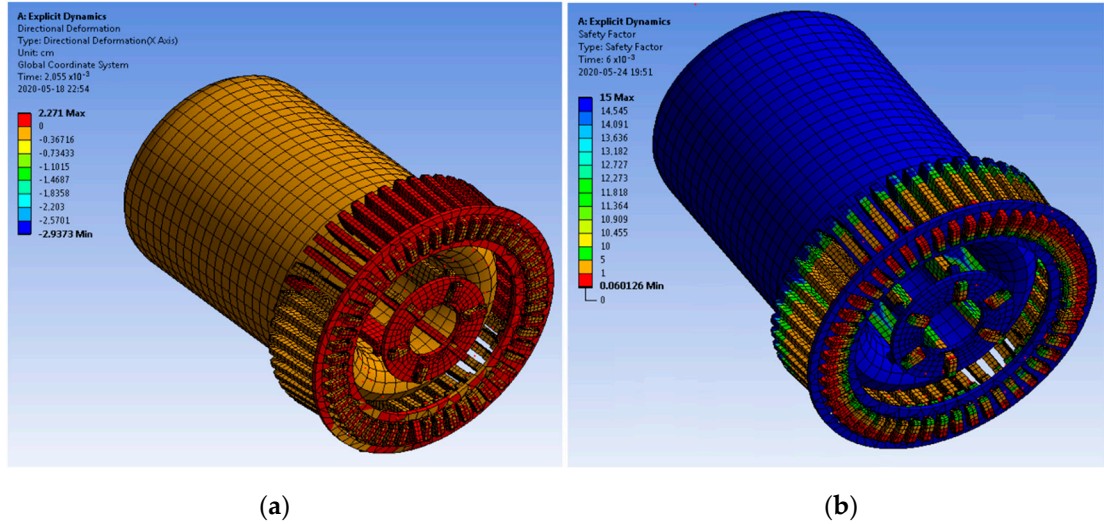

(**a**)  (**b**)

**Figure 9.** View of the tulip contact analyzed in ANSYS: (**a**) View of the direct values of contact deformation and (**b**) the safety factor.

In a further part of the analysis, individual elements of the tested system were discussed. In Figure 10, the discretization grid (mesh) and results of deformation analysis can be seen for lamellas. the mesh was compacted with fragments, requiring more detailed analysis. the views of the lamellas show that the places where they are attached in the crowns are exposed to high overload. In these places, deformation of the material occurs very often. the ANSYS program showed red loose dots in the output. In this way, it represents the crumbled lamella or crown material. Chips of this type occur as a result of two elements hitting each other with significant force. During further work on a given

lamella, the connection between the lamella and the crown should be rebuilt by reducing friction and optimizing the distribution of forces and movement of the element.

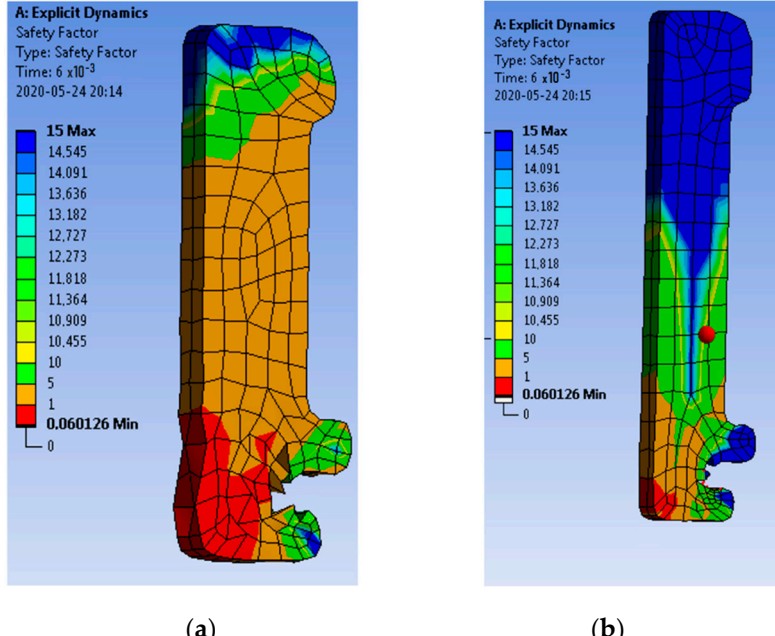

|            (a)            |            (b)            |

**Figure 10.** View of the lamellas used in the tulip contact: (**a**) Small (external) lamella in the deformation value view and (**b**) large (internal) lamella in the deformation view.

Crowns were the next element observed to be more susceptible to deformation. Figure 11 shows that the crowns carry high mechanical loads through them. It should be remembered that the crowns rest on the lamellas, which, in turn, are hit by the moving part of the contact. In a later phase of movement, the lamellas deviate from the contact axis and, being fixed in the lower part, exert torque on the crown. This causes the force of the crown to become deformed. In the small crown, the principle of movement and the type of forces are the same, but the direction of the slat deflection is the opposite, due to its setting. When considering the construction of the crown, it should be remembered that it must support the lamellas longitudinally in relation to the contact axis and prevent them from deflecting sideways.

*6.4. Results of Motion Dynamics—Variant I*

All of the above-mentioned results were read from colored scales next to the elements of the tulip contact system. These were also presented graphically as the course of the functions of force, moment, and energy. Color maps of forces marked on the contact elements facilitate quick analyses of the tested value. an hourglass effect can be seen in Figure 12. In order to analyze all of the movement, it is significant to refer to the graphs below Figures 13 and 14. In the first graph—Figure 15—the course of kinetic energy, internal energy, and the hourglass effect (Hourglass Energy) is shown. It can be noticed that from the first moment of time, the input has the highest kinetic energy. This proves that the moving part of the contact was activated at the beginning of the simulation. With the flow of time in the motion analysis, it can be clearly seen that the kinetic energy value is decreasing. This is the moving part of the contact that moves downwards. As a result, the distance between the elements decreases. In accordance with the formula for kinetic energy, its value decreases. In the same diagram, the course of the internal energy of the system can be viewed. This energy increases as the simulation runs, reaching the maximum values just after the movable contact hits all of the lamellas. the red line in the same figure shows the hourglass effect. This is a false deformation of the finite element mesh

resulting from the stimulation of the freedom degrees of zero energy. It usually manifests itself in a cluster of zigzag lines or shapes in which the individual elements have strongly deformed edges.

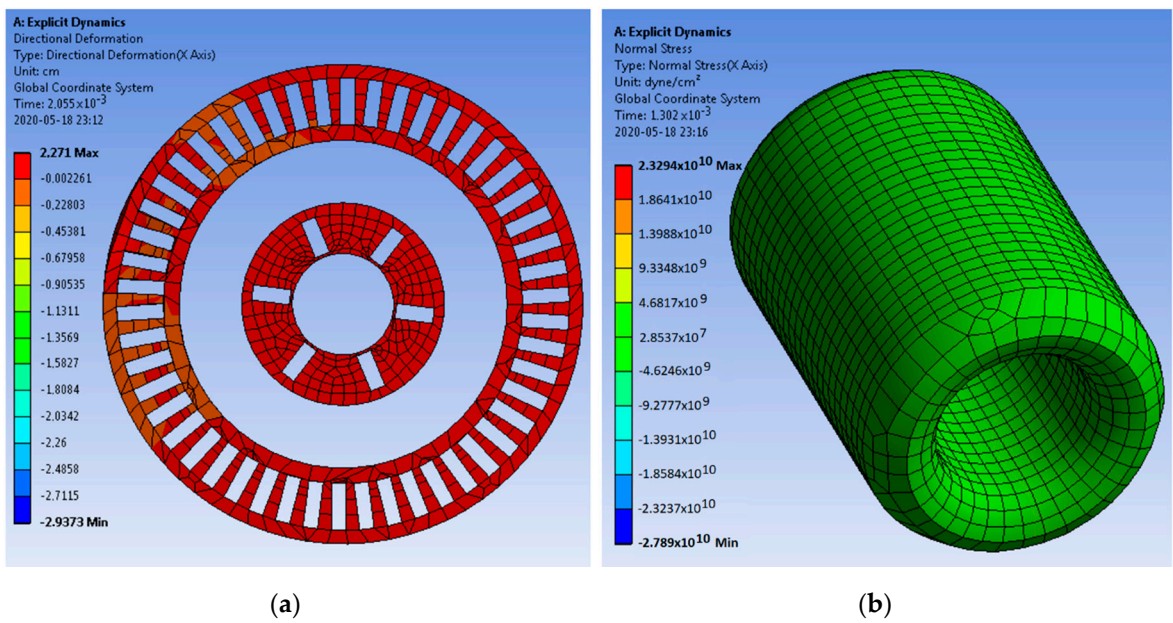

(**a**)                                     (**b**)

**Figure 11.** Crown analysis: (**a**) View of the lamella fastening crowns and (**b**) view of the movable part of the contact.

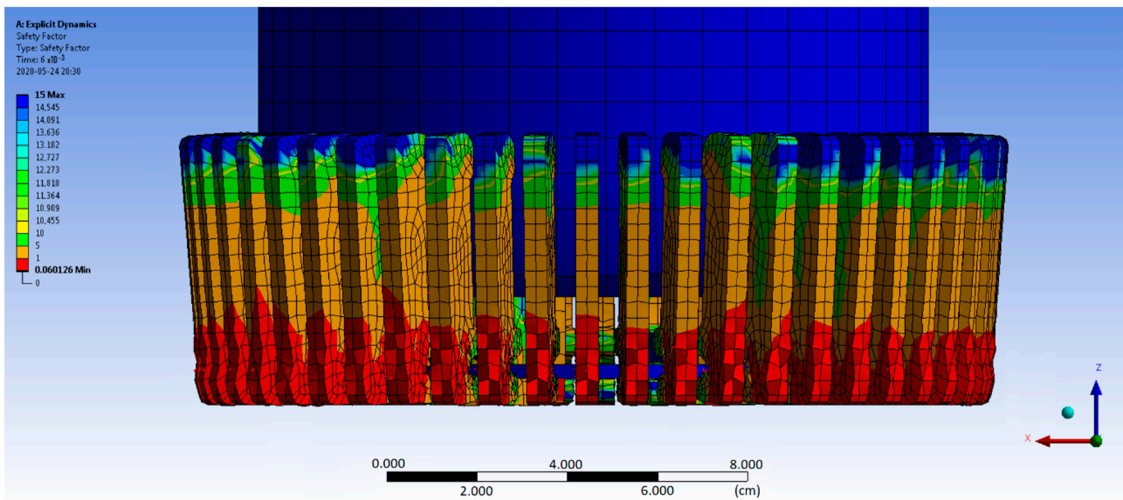

**Figure 12.** View showing the hourglass effect.

Therefore, the red line in Figure 15a runs below the value of the internal energy of the contact until the collision of the movable part and the small lamellas. This proves that this effect is of negligible value. However, after hitting the small lamellas, this value increases and reaches its maximum as soon as the motion of the moving part is finished and the small lamellas stop keying. During the impact, the small lamellas detach from the axis of the system, and thanks to the pressing forces, they hit the external part of the contact and begin to press on its surface. De-keying and keying cause compressive stresses in the rear parts of the lamellas, which, for a given discretization grid, result in the effect of glass-gluing. To prevent this effect in subsequent iterations, the discretization mesh should be increased locally. This will directly contribute to increasing the density of calculations and reduce the phenomenon of false deformation of the material. the next step was to increase the pressure of the lamella. This can reduce the angle of the throw and the keying itself.

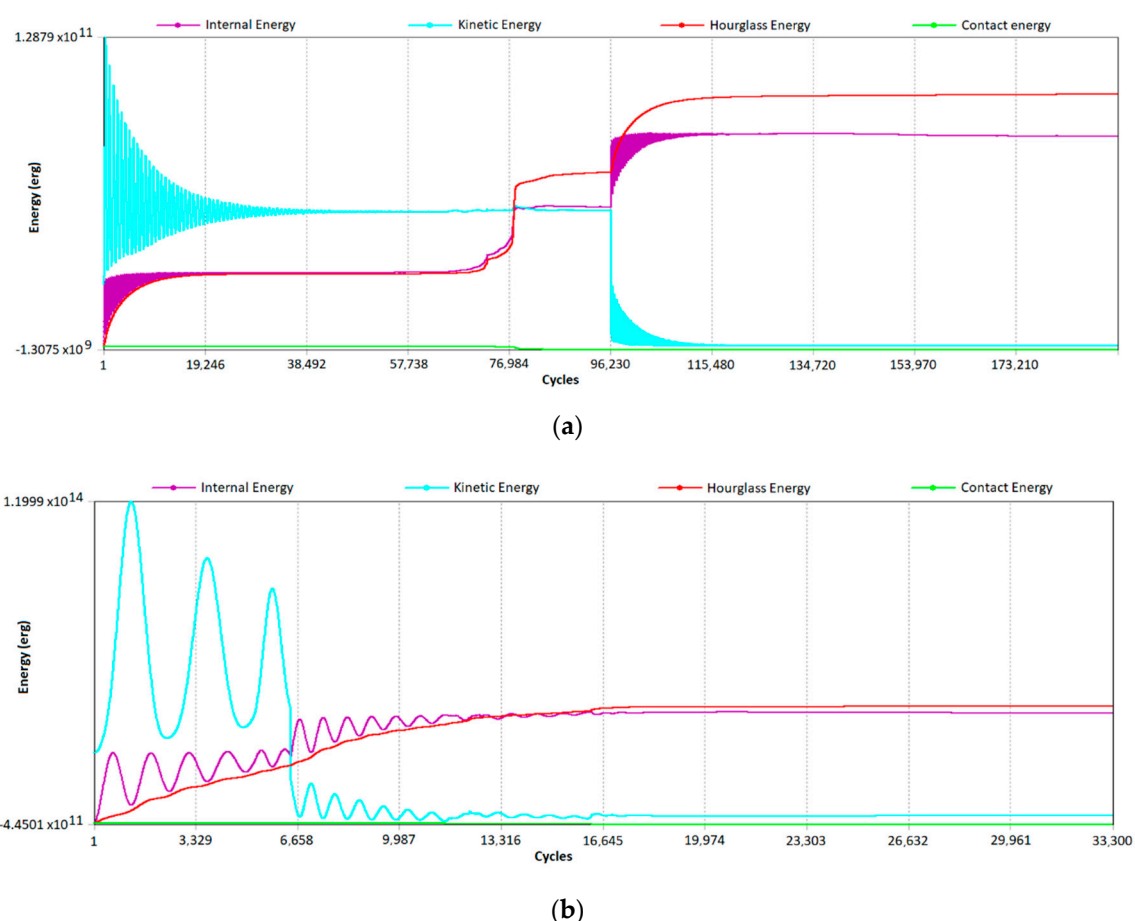

(a)

(b)

**Figure 13.** Diagrams of the tulip contact motion analysis: (**a**) Energy summary—normal operation and (**b**) energy summary—contact destruction.

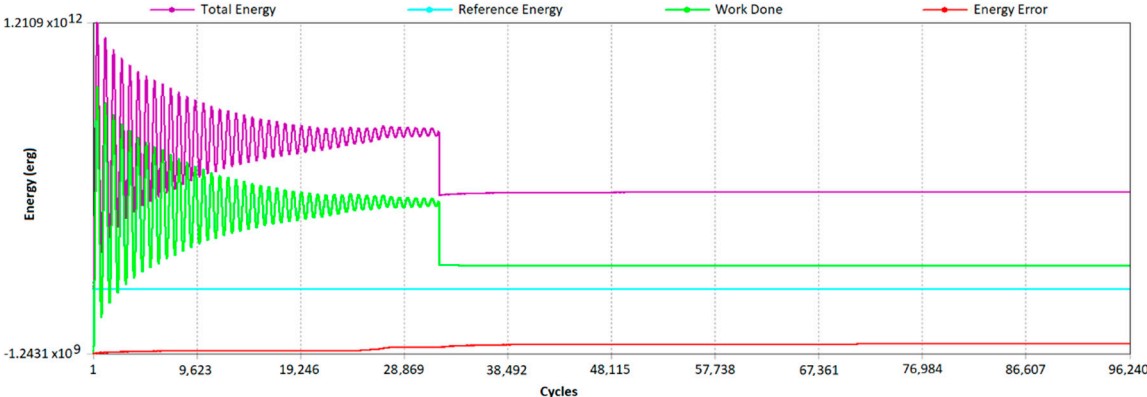

**Figure 14.** Graphs produced from the analysis of the motion of the tulip contact, energy consumption, and energy conservation.

The graph in Figure 15b shows the values of two parameters: the total energy and work. It can be seen that energy and work follow the same course. This means that the calculations are consistent with real conditions. With the passage of motion time analysis, a decrease in the value of both energies can be noticed, which is caused by the movement of the contact. After moving the movable part of the contact to a lower position, there is still a moment of vibration on the small blades. the energy and work drop to low values. Note that the error value increased at the end of the move. Nevertheless, it was still significantly below the assumed error limit.

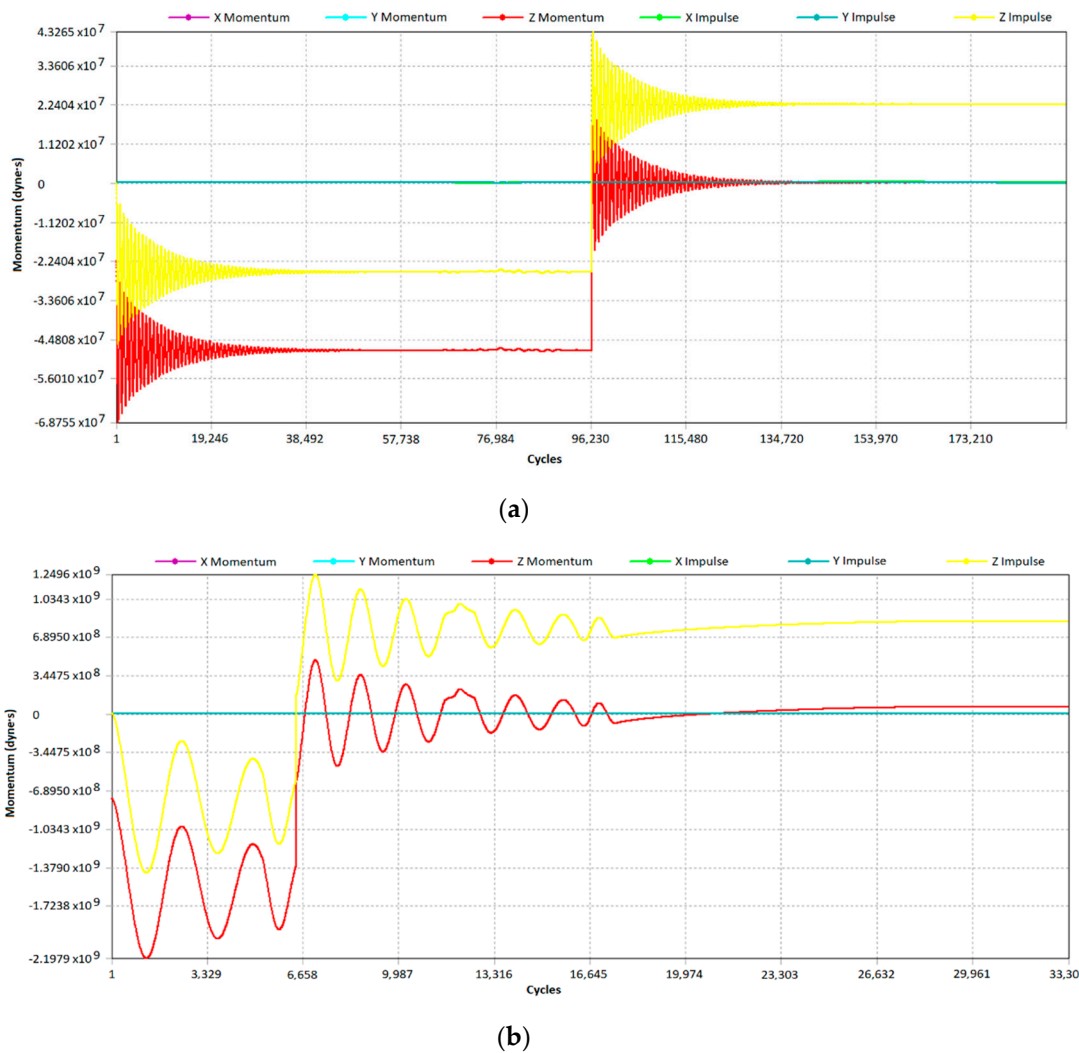

**Figure 15.** Graphs produced from the analysis of the motion of a tulip contact, presenting a summary of moments: (**a**) Normal operation and (**b**) contact destruction.

The main torque and energy impulse runs along the axis of the tulip contact. It can be noticed that the greatest moment exists at the time of the maximum velocity and force of the moving part of the contact. Then, this value decreases as the elements collide with each other. the moment value then returns to its original value. This is due to the fact that the movement of the cylindrical part of the contact is negative in relation to the main reference system. the lamellas, on the other hand, deviate in positive directions. At the time of reading the lamellas, an increase in the moment was noticed, which also, after a while, began to stretch across the frictional forces and stresses inside the elements.

*6.5. Analysis of the Tulip Contact Motion Dynamics—Variant II*

The second variant revolved around an analysis of the nearly exact contact system, but the radius of the lamellas with the moving part of the contact was modified. Figure 16 shows the difference in the radius of the lamellas and therefore, the change in contact parts' geometry.

Figures 16 and 17 show the modified lamellas. Figure 17 additionally shows the results of the analysis. Two fundamental differences were noted. In the upper part of the lamella, the force distribution, namely the yellow line representing the value 10, has a completely different course. the movable parts of the contact strike the lamellas at different angles in both cases. In the first case, when the lamella radius is greater, the point of impact is closer to the edge of the internal contact axis. This translates into a lower deflection force of the lamella from the moving contact compared

to the second case. the reduction of the generating radius in the second case increased the friction during the impact, which implied the creation of higher force values in the entire lamella. the yellow lines of force values reflect how the direct impact is distributed to the stresses inside the material. It was observed that a larger radius also translated into greater forces in the upper part of the lamella. the stress distribution in the middle is identical in both cases. However, the lower part of the lamellas, which was fixed in the crown, shows considerable damage in the reduced radius version. the greater recoil force of the lamellas resulted in greater stresses of the lamellas in relation to the crown, and thus deformation of the material.

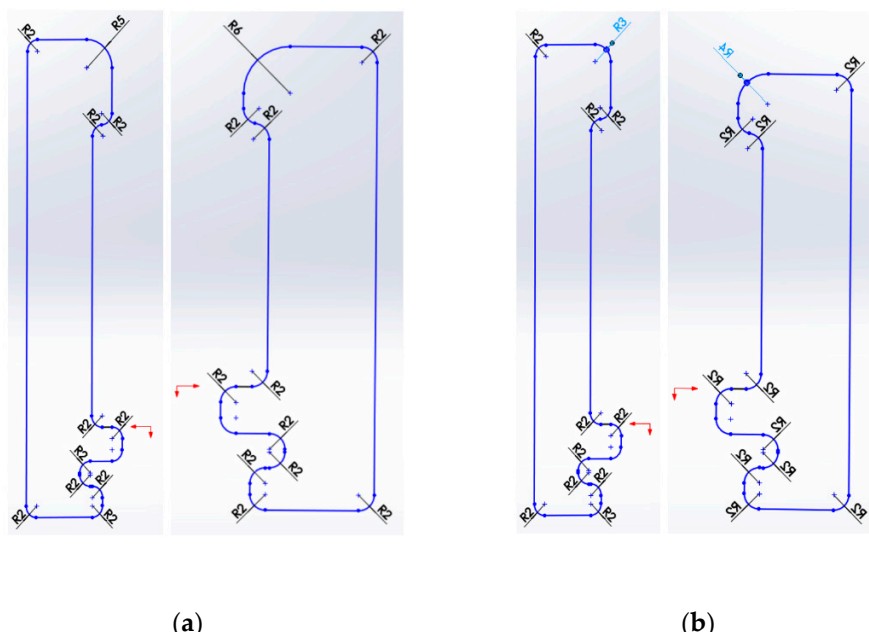

(**a**)          (**b**)

**Figure 16.** Lamella views: (**a**) First variant and (**b**) second variant.

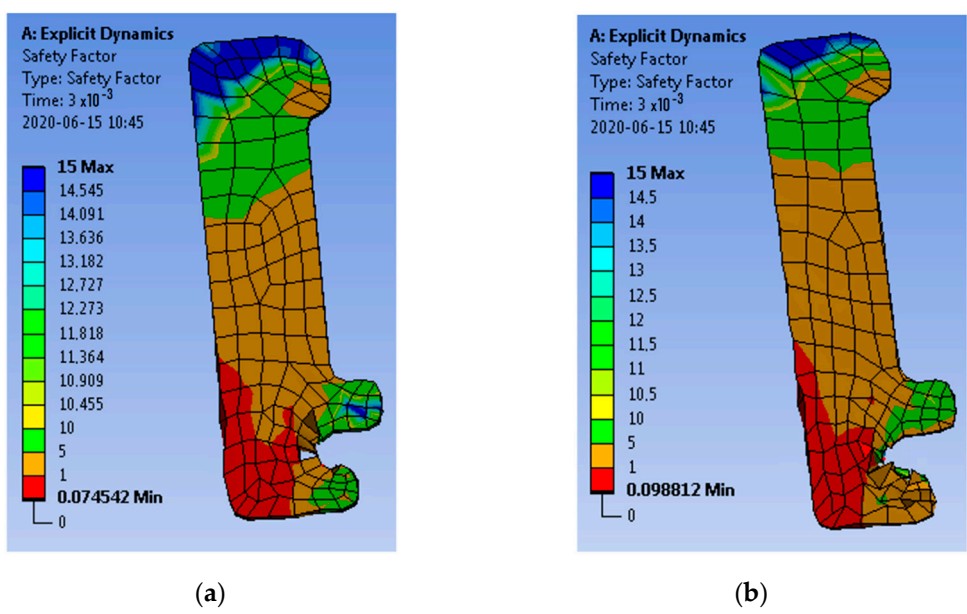

(**a**)          (**b**)

**Figure 17.** View of small lamellas: (**a**) Old lamellar radius R6—Variant I and (**b**) new lamellar radius R4—Variant II.

Figure 18 shows large lamellas with the same modification as described above. In these lamellas, it is harder to notice significant deformations than for smaller lamellas. This is due to the fact that the total length of the lamella is greater compared to the small lamellar, which was translated into greater plasticity of the entire element.

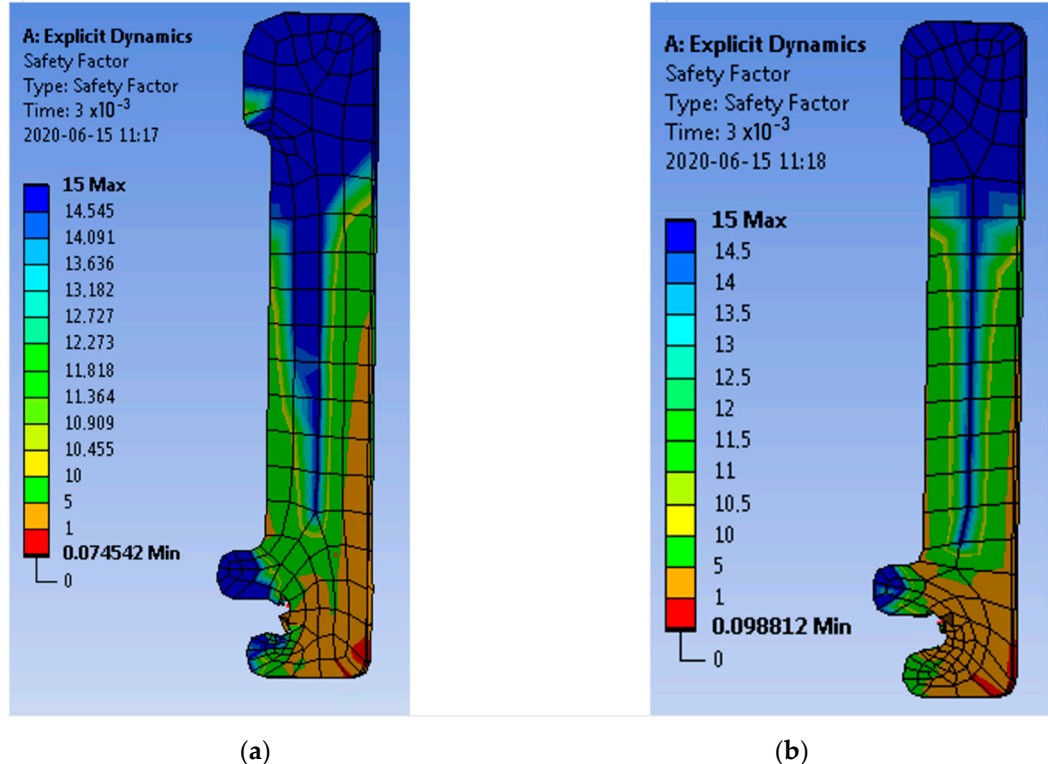

(**a**)                                              (**b**)

**Figure 18.** View of large lamellas: (**a**) Old lamellar radius R5—Variant I and (**b**) new lamellar radius R3—Variant II.

The results show a comparison of the values of the first analysis and the analysis performed after the modification of the lamella. Figure 18 shows the velocity of deflection of the lamella in each plane, and the values are expressed in cm/s. It was noticed that the velocity values of the lamellas after modifying their contact part with the moving contact were lower. Comparative charts of the velocity in a given plane before and after the modification clearly show the impact of changing the structure of the contact surface and impact on the behavior of the lamella. It turns out that after the modification, the lamellas move slower, with smaller oscillations, and with slower velocities in each of the planes.

The energy and torque values presented in Figure 19 show the values of the above-mentioned parameters in relation to the system with rebuilt lamellas. the above diagram should be read in combination with Figures 13a and 15a. Then, it can be seen that the energy values in the second analysis were much higher. It was also noted that a much lower hourglass effect was achieved. Additionally, the energy values reached higher values, whilst their shape and form remained unchanged. Figures 19 and 20 are presented below.

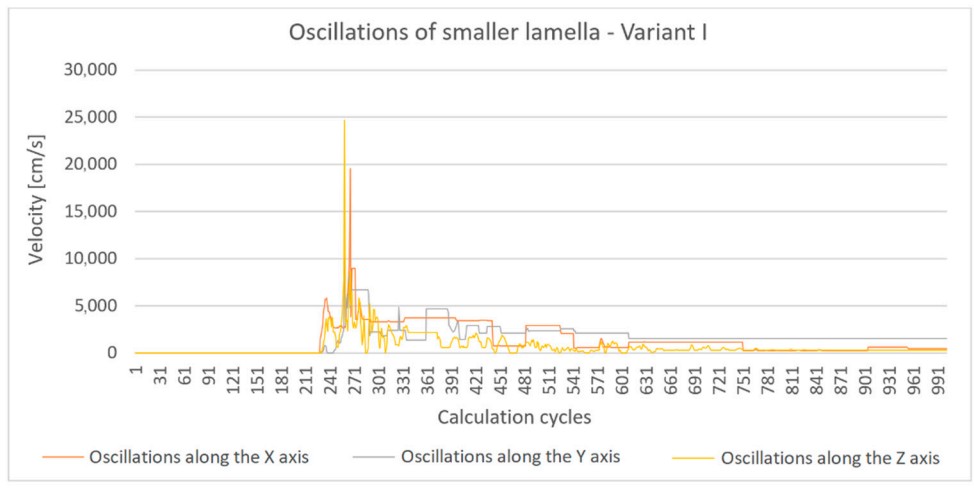

(**a**)

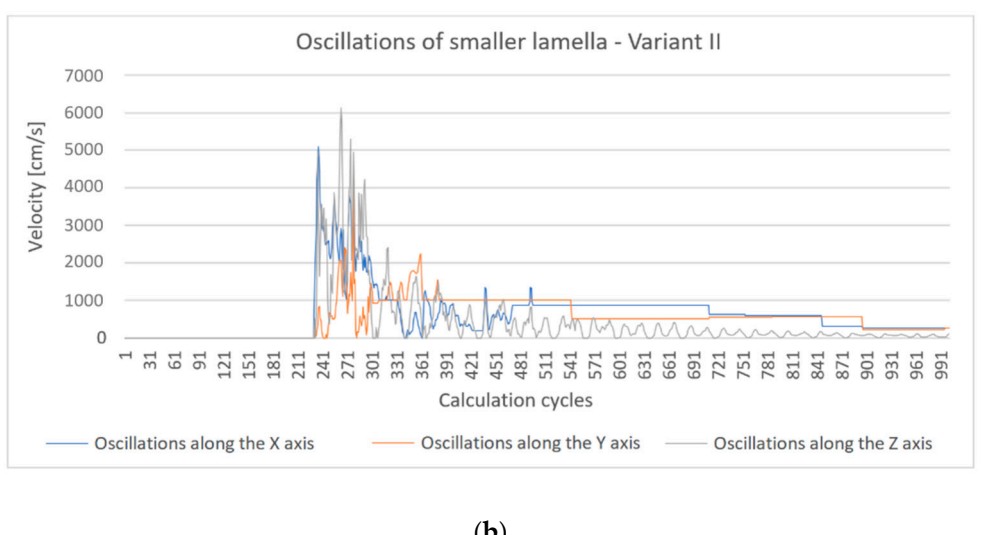

(**b**)

**Figure 19.** View of the small lamellas' oscillations before and after modification in each plane: (**a**) Variant I and (**b**) Variant II.

### 6.6. Defects Study in Motion Dynamics Analysis

During the preparation of the dynamic motion analysis, a number of simulations were performed. In the initial phase of developing the tested model, the execution of a three-dimensional representation in the SolidWorks environment was conducted and the model was prepared in the ANSYS software. In the course of releasing parameters from constraints, replacing forces, assigning contacts, and determining the friction area, their coefficients' influence of various weak and weakened points on the correct operation of the system was observed. These types of observations, despite the fact that they resulted from complete randomness, uncovered significant assets. Thanks to such analyzes, which type of failure of a single element may affect the operation and generate faults and aging correlated with certain parameters of the apparatus was observed.

Figure 21 shows the damaged crown of the small lamellas. This damage occurred as a result of several factors. One of them was the wrong kind of contact between the moving part of the lamellas and the large crown. This resulted in too much torque exerting pressure on the crown and thus in its deformation while deflecting the lamellas from the axis of the camera. In real conditions, this type of case may occur when the contact is not maintained. This can result in jamming of the moving parts on

the contact parts. Another reason was the insufficient forces pressing the lamellas against the contact axis. These forces were due to the release of the constraints against the springs surrounding the contact. As a result, when the movable part hit the lamellas, the crowns were significantly deformed. In extreme situations, the lamellas tear the crown apart and fall outside their area and the contact is completely destroyed. Such a situation is shown in Figures 22 and 23.

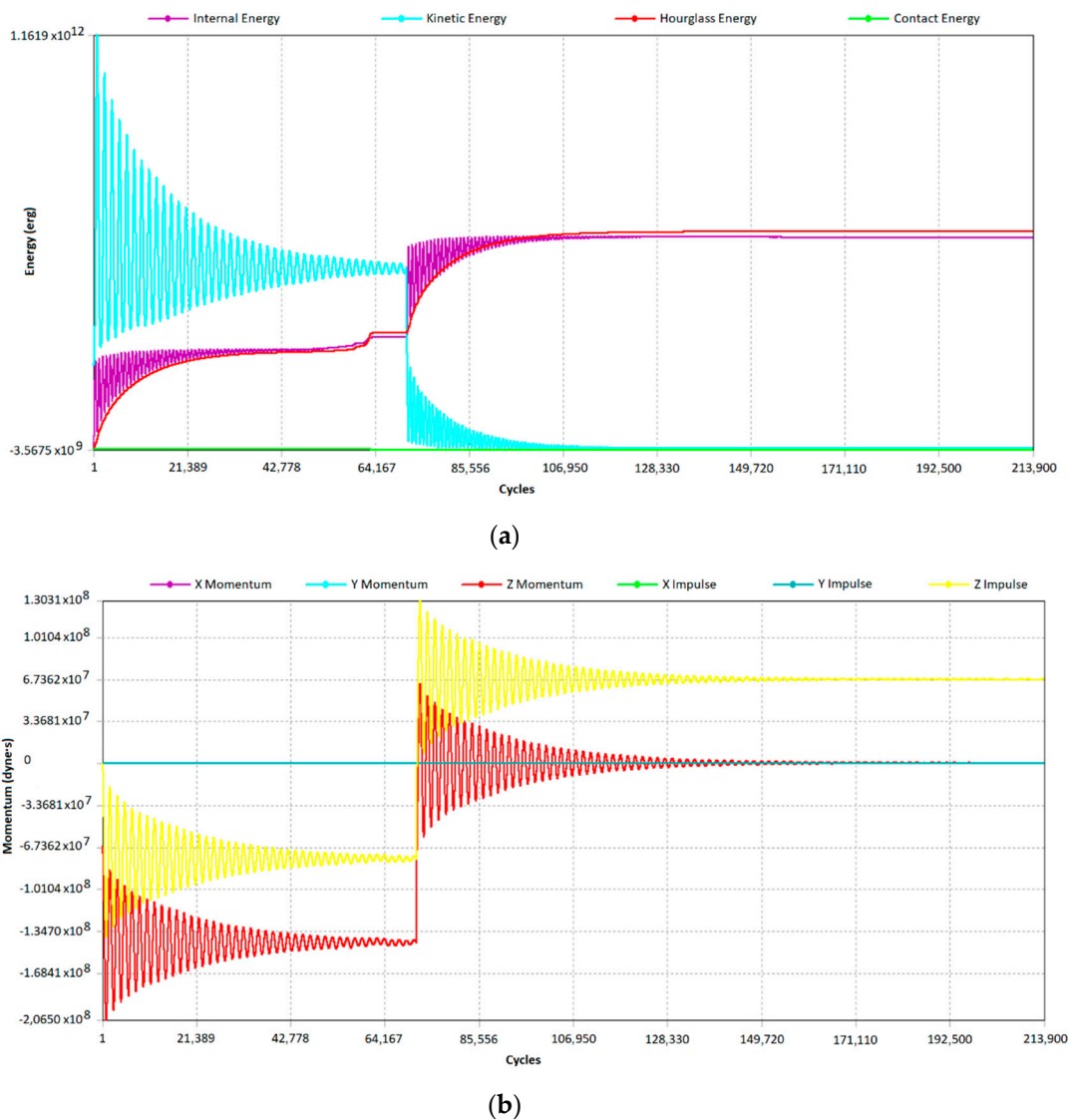

**Figure 20.** Tulip contact motion analysis charts: (**a**) Sum of energy—Variant II, and (**b**) torque value—Variant II.

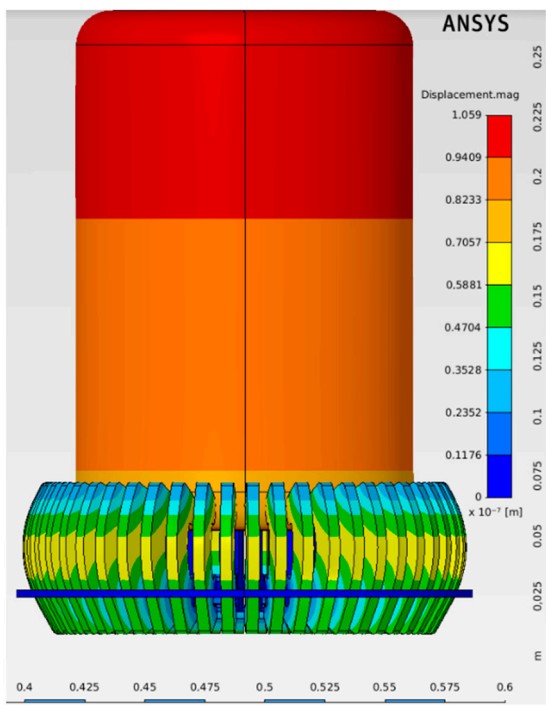

**Figure 21.** Deformation of small lamellas.

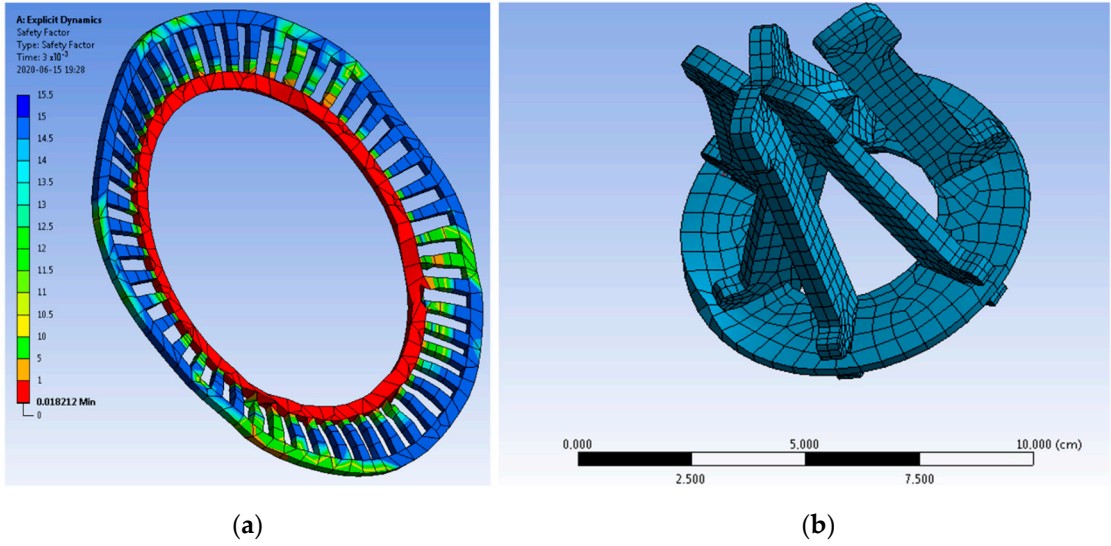

(**a**)                              (**b**)

**Figure 22.** Deformation of the crowns: (**a**) Outer crown and (**b**) inner crown.

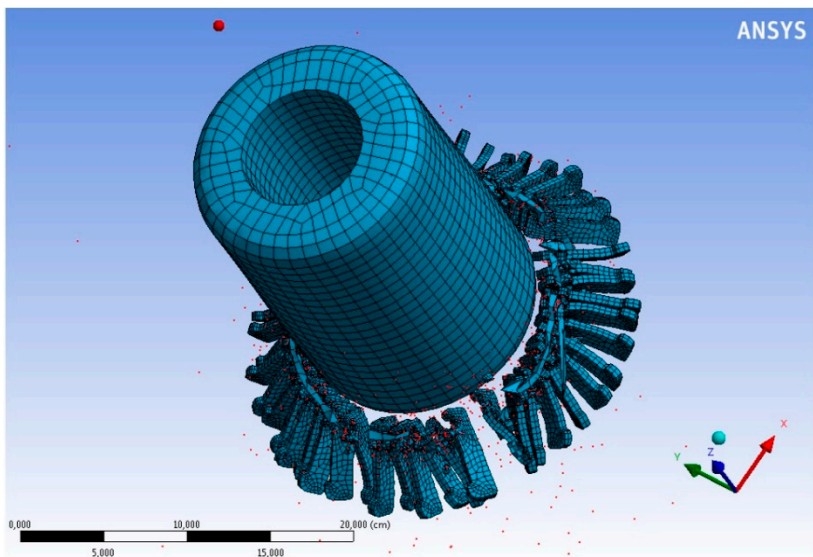

**Figure 23.** Destruction of the entire contact system.

## 7. Electrical Simulations of the Tulip Contact System

### 7.1. Electric Field Distribution in a Tulip Contact System

The first considered case concerns the contact in the open position (NO). the ground potential *V* was equal to 0 V and was assigned to all lamellas and both crowns. the movable part of the contact was assigned a potential of V equal to 110 kV. the color maps below show the distribution of the potential and the electric field. Thanks to these values, the points that will first form the opening of the plasma channel and the burning of the electric arc were observed. This information is extremely important when designing a contact system.

Figure 24a above shows the electric potential distribution around a contact. It is worth noting that a given distribution is symmetrical along the "z" axis. Such data provide information about evenly distributed potential, which translates into switching parameters. Each of the contact elements at the same level, with the same distance from the movable part of the contact, are at the same voltage level. During the operation of the tulip contact, this will cause an even distribution of the electric field, which will have a direct impact on the formation of sheaf and partial discharge phenomena. Uniformity of the field distribution ensures greater switching properties and a longer exploitation time for the contact system.

Considering the field phenomena further, it should be mentioned how the results will be presented. Figure 24b above shows the plane inside the computational cuboid. the plane creates an intersection through the model and represents the distribution of the studied value over the plane. the location of a given surface is freely defined.

Figures 25 and 26 present the distribution of the electric field with the exact space between the large lamellas and the moving part of the contact. In line with the assumptions, it was noticed that the greatest electric field gradient occurred between elements with different potential and close to each other considering sharp edges of the elements. It was observed that the field distribution between the right and left side of the contact was symmetrical. This proves the central axial location of the elements in relation to each other. Maintaining the alignment brings not only mechanical benefits, but also visibility during the movement of the contact and electric potential in both open and closed positions. a uniform field distribution in the open position of the electrical apparatus results in the appearance of partial discharges. Thanks to this, it is known which sites of the contact will be constantly exposed to this type of discharge and consequently, to the appearance of a temperature source and slow burning of the surface in a given place. However, from the moment of starting

the contact closing procedure, the field gradient begins to increase rapidly until the formation of a plasma channel, ignition of the electric arc, and the collision of lamellas. the uniformity of the electric field also produces an even distribution of electrodynamic forces, in accordance with the Biot–Savart and Ampere standard laws.

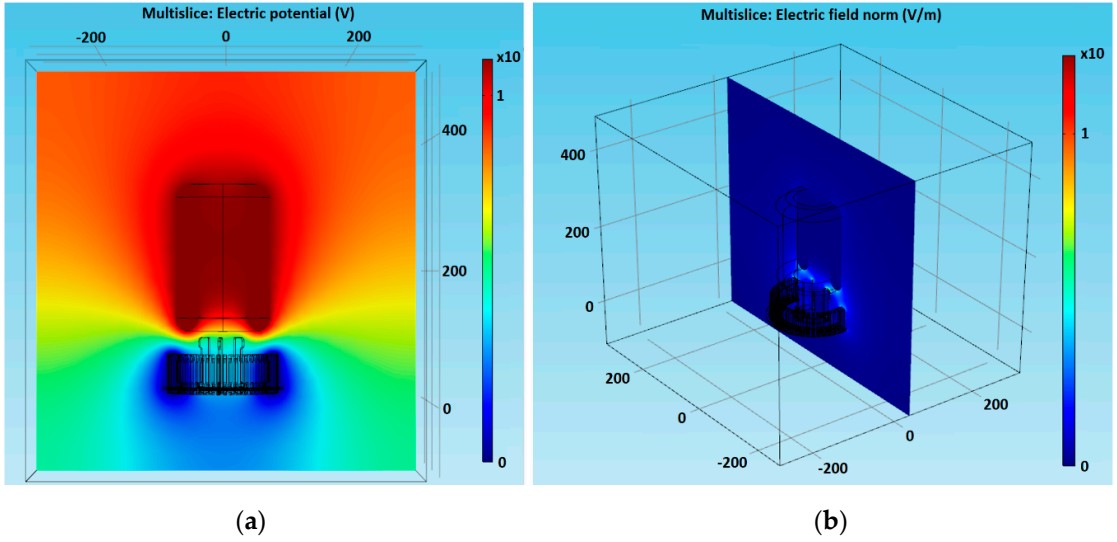

                (**a**)                                                   (**b**)

**Figure 24.** Electric potential distribution: (**a**) Potential distribution around the tulip contact (*V/V*) and (**b**) plane showing the calculated values.

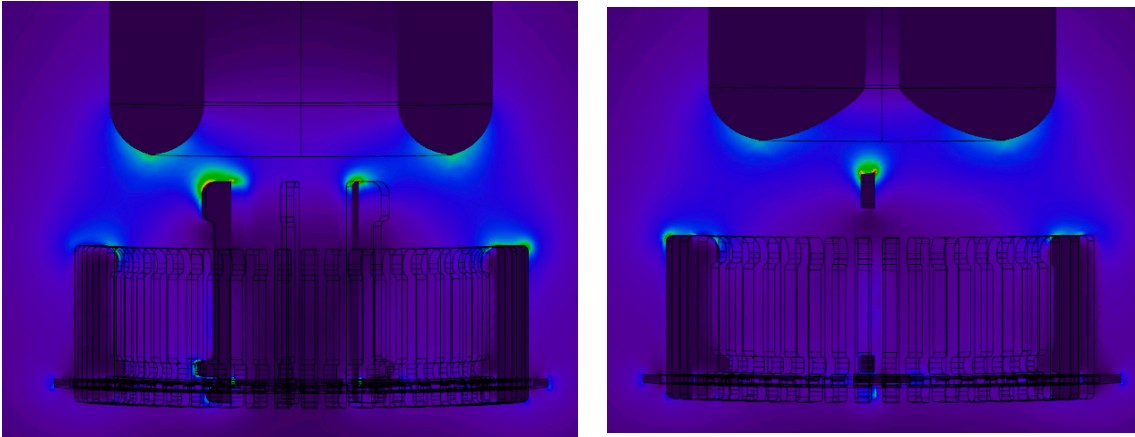

**Figure 25.** Electric field intensity distribution of large lamellas (*V/m*).

The movement of the moving part of the contact towards the lamellas causes a greater intensity of the electric field. If any of the lamellas have a different distance or angle than the others in relation to the working part of the contact, this would cause electrodynamic forces to appear in the contact lamellas at different times. This would translate into a resultant force different from zero acting transversely to the direction of movement of the working part. the appearance of this type of force can cause the moving part to deflect by several degrees. Small lamellas may hit at the wrong angle, and the appearance of electrodynamic forces may thus differ in time for distinct lamellas. Repeated connections with this type of fault will lead to destruction of the contact.

The electric field in switching systems can be managed in a number of ways. One of them is the use of field diffusing materials. Another way is to use contact elements with an appropriate geometry, so that the field gradient is highest in the right places.

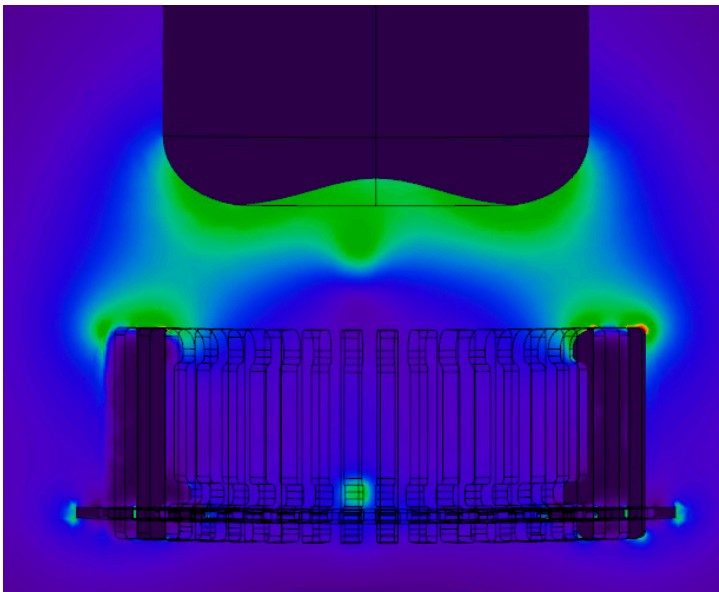

**Figure 26.** Electric field intensity distribution of large lamellas (*V/m*).

### 7.2. Parametric Analysis of the Electric Field

The further part of the electrical analysis was carried out in several steps determining the position of the movable part of the contact in relation to the lamellas. In this approach, the natural work of the tulip contact was studied. During each position, the field distribution between two large lamellas and between several small lamellas was tested. the analysis was conducted as follows. a line was drawn between the lamellas, indicating between which elements the measurements were made. the positions of both lines are shown in Figure 27 below. the obtained results are presented graphically on the charts in Figures 28–30. the results of individual analyzes were compared and discussed.

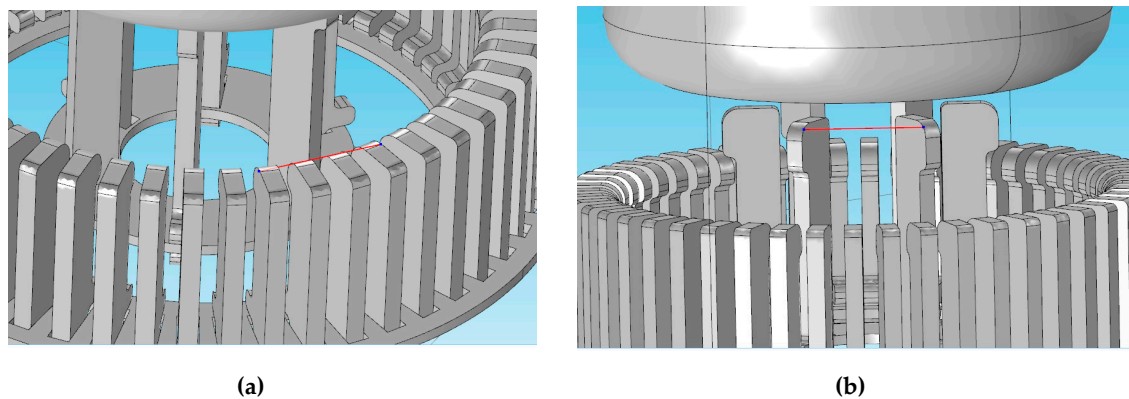

(**a**) (**b**)

**Figure 27.** View of the measurement lines: (**a**) Small lamellas and (**b**) large lamellas.

Figure 27 shows the distribution of the field along the measurement lines. According to the legend, the blue line represents the distribution of the electric field. It was noticed that the distribution character was repeatable and symmetrical in relation to the adjacent lamellas. This proves the uniformity of the field distribution on the individual contact elements at the same distance from the movable part of the contact. the moving part is a hollow cylinder with chamfered edges. the purpose of chamfering is not only to reduce the friction and obtain an appropriate angle between the connecting surfaces, but also to reduce the sharp edges and limit the electric field gradient. This leads to an even distribution and an increase in the dielectric strength of the system. There were also differences in potential between the edges of the same lamella. However, this does not indicate an actual difference in potential at

the edge of the element, but highlights how to measure a given parameter. the measurement line is a chord between several elements. Therefore, the normal distance on both sides of the lamella to the chord is different, as observed in potential differences.

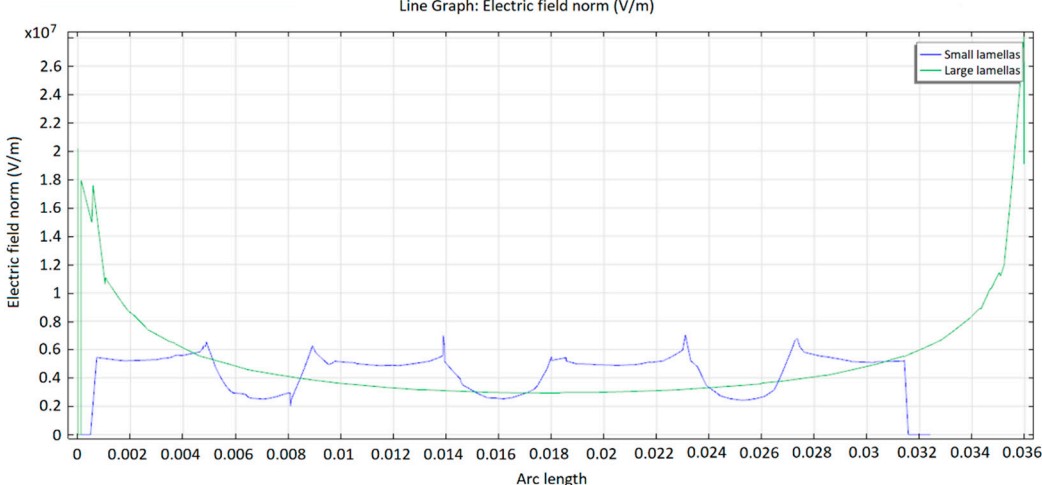

**Figure 28.** Results of the electric field distribution along the measurement lines.

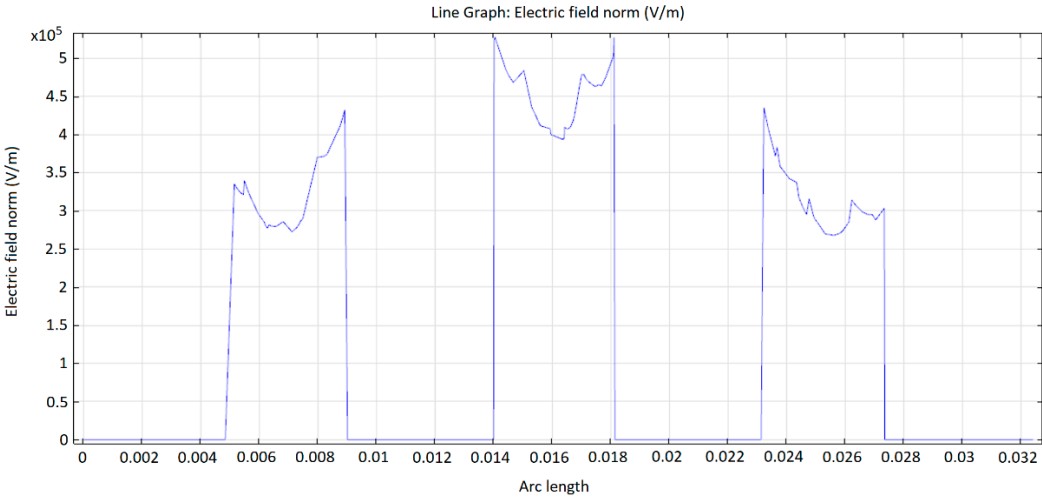

**Figure 29.** Results of the electric field distribution along the measurement lines for small lamellas.

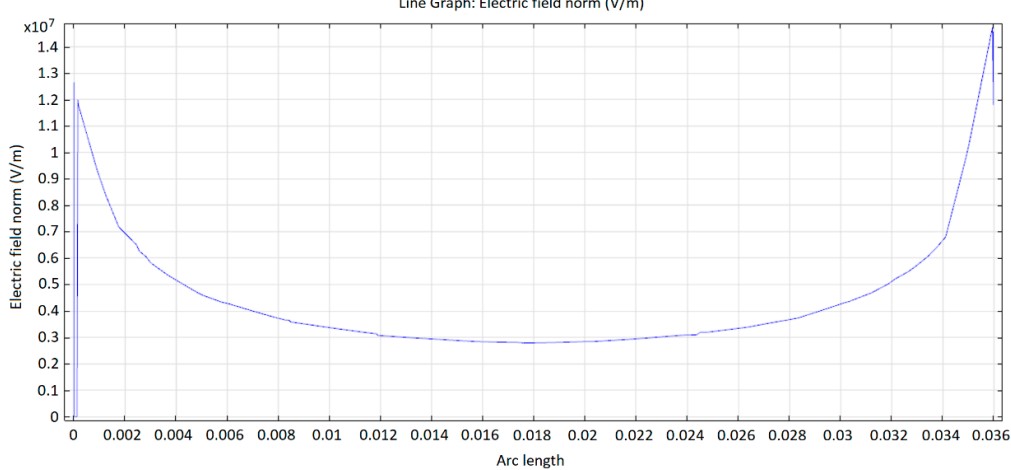

**Figure 30.** Results of the electric field distribution along the measurement lines for large lamellas.

In Figures 29 and 30, the field distribution can be observed with the moving part displaced by 5 mm towards the lamellas. Figure 30 represents the distribution of the electric field, along the measurement line placed next to the large lamellas. a decrease in the intensity value can be observed. This is due to the proximity of the moving element and the reduction of the field distribution gradient. There was also a reduction in the difference between the value on the left side of the chart and on the right side. the lower difference is also due to the more uniform field. the same applies to the measuring line next to the small lamellas which were presented in Figure 29 above.

## 8. Validation of the Procured Simulations

The validation was carried out on a tulip contact made in the short-circuit laboratory of the Warsaw University of Technology. It was not a construction like in the presented simulations. Nevertheless, it exactly reflects the nature of physical phenomena witnessed in simulations.

In practical solutions, the inner surface of the contact point differs significantly from a completely flat one. Depending on the solution, the active part may only be located in the closest vicinity of the pressure spring. In other words, there may be distinct grooves to prevent welding of the contacts. To determine the initial spacing between the contacts, it is suggested that a limiting ring is used (Figure 31). the upward movement of the contact should also be blocked in the first moment after hitting the movable contact—a bumper is used for this purpose.

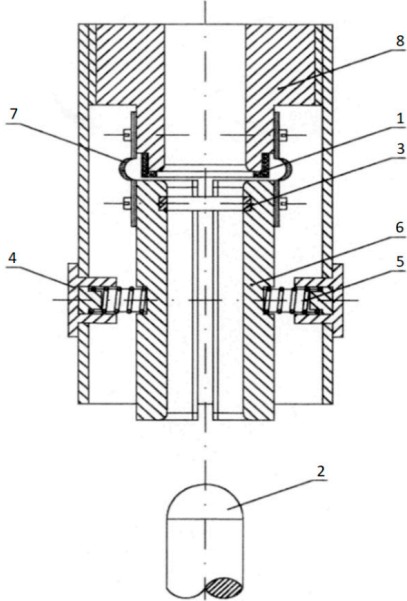

**Figure 31.** Tulip contact: 1—impactor; 2—movable contact with an arcing tip; 3—limiting ring; 4—compression spring guide; 5—compression spring; 6—pins; 7—flexible connection; 8—body of fixed contact.

The operation of a high-voltage tulip contact in a dynamic system was analyzed on the test stand, taking into account the welding currents. the tests were carried out with the use of a short-circuit system. the laboratory stand is presented in Figure 32 below.

On the oscillograms shown in Figure 33, the following negative phenomena occurring in the contacts during closing were observed. First of all, a bounce phenomenon was observed on the graph showing the puss—the point of contact of the contacts—Figure 33a. It is visible in the form of oscillations. It is a disadvantageous phenomenon because it causes the formation of an electric arc, which in turn deteriorates the electrical properties by gradual burning of the contacts. In order to prevent this phenomenon, the contact pressure force should be changed.

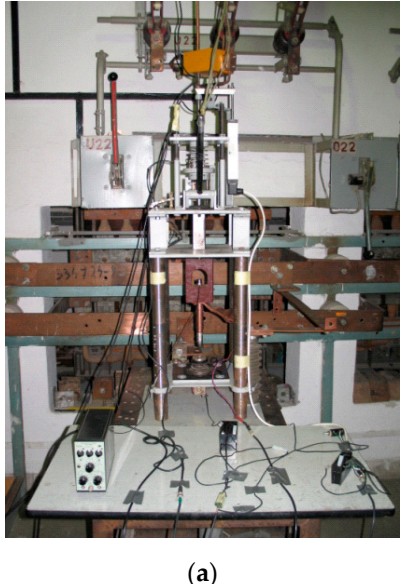

(**a**)

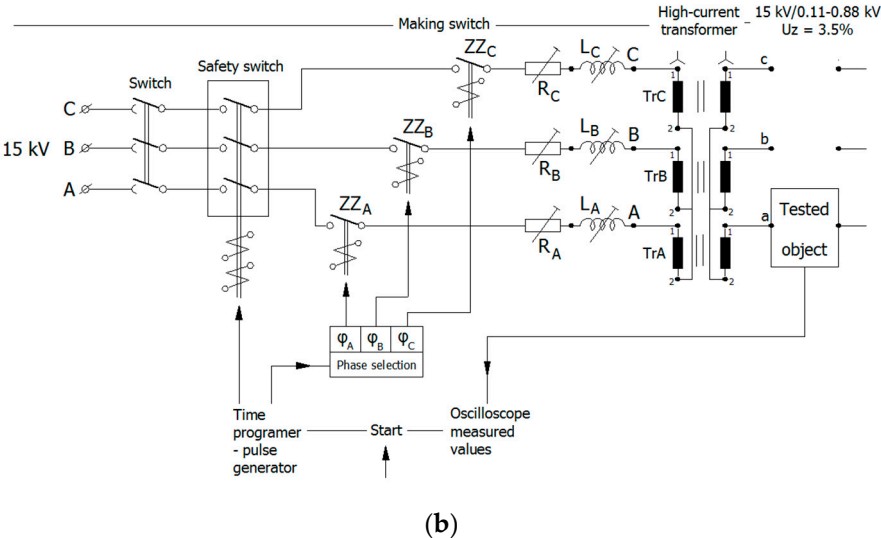

(**b**)

**Figure 32.** Laboratory setup for simulation validation: (**a**) Laboratory stand and (**b**) short-circuit system in the Short Circuit Laboratory, Electrical Faculty, Warsaw University of Technology.

Another disadvantageous phenomenon is the friction force between the contact elements. It is perfectly visible, also in the form of oscillations in the course of acceleration and speed. the last observed phenomenon is the welding of contacts. Welding the contacts together leads to the necessity to increase the value of the force needed to open the contacts. In this case, there is also the possibility of an electric arc appearing between the contacts. On the basis of the tests performed and the recorded velocity waveforms, it was possible to determine the speed of the contacts' convergence. Full validation was executed in the Warsaw University of Technology Laboratory.

Figure 34 shows the characteristics of the movable contact velocity for two cases: Without bounce and with movable contact bounce. Characteristics derived from the simulation were compared to those which had been captured during laboratory tests.

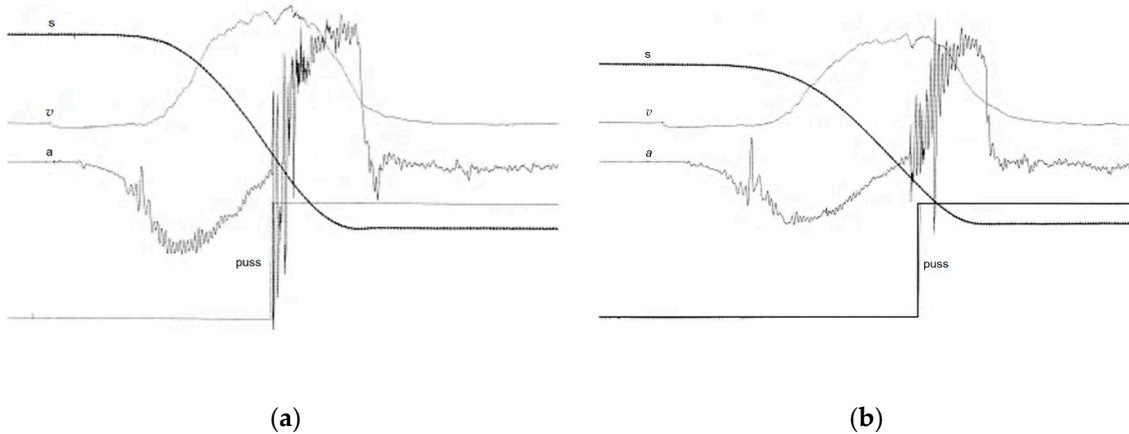

<div align="center">(<b>a</b>)       (<b>b</b>)</div>

**Figure 33.** Exemplary laboratory results of the tested tulip contact system: *v*—speed, *a*—acceleration, *s*—path, and puss—characteristic that indicated contact between system elements. (**a**) With bounce and (**b**) without bounce.

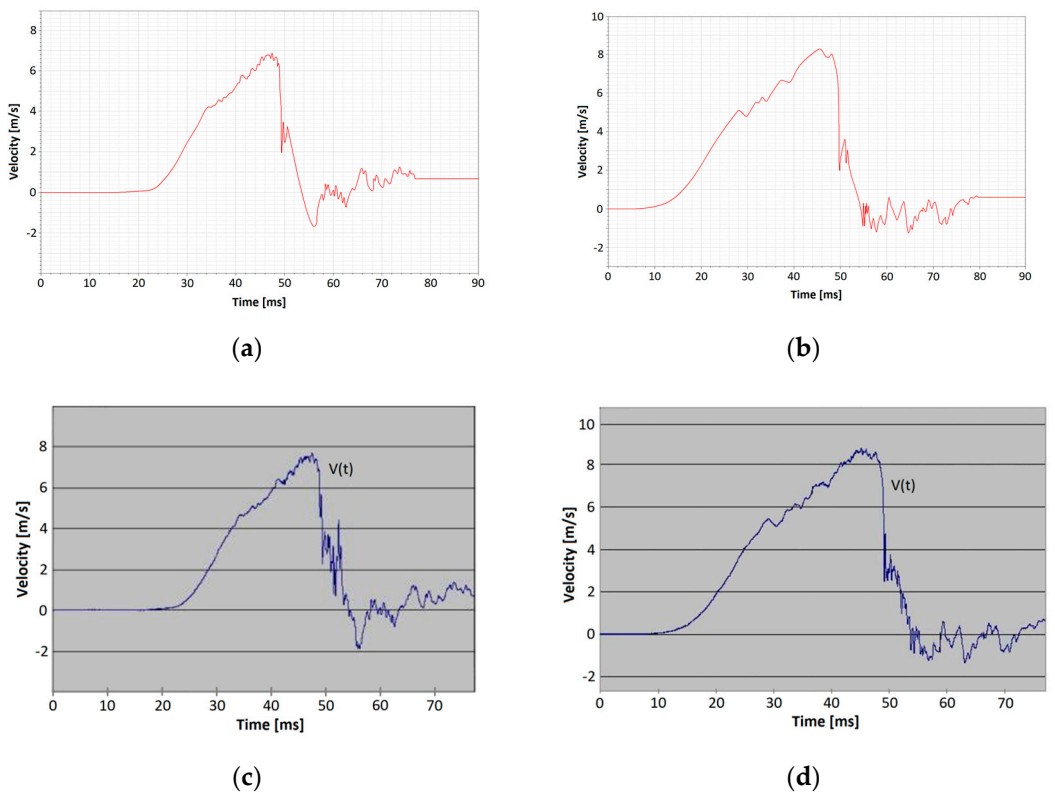

<div align="center">(<b>a</b>)       (<b>b</b>)</div>

<div align="center">(<b>c</b>)       (<b>d</b>)</div>

**Figure 34.** Juxtaposition of characteristics derived from the simulation in comparison to those captured during laboratory tests. Ansys characteristics (**a**) without movable contact bounce and (**b**) with movable contact bounce. Laboratory test characteristics (**c**) without movable contact bounce and (**d**) with movable contact bounce.

## 9. Summary

Each described case of analysis should be considered individually, remembering that a contact is a system of many elements cooperating with each other. By improving one factor, the optimal values of the other may deteriorate. When defining the project, which is the design of the switching system, it is essential to define the conditions, purpose, and natural working environment of a given electrical apparatus. In the considered cases, the change in the geometry of the contact surface and

the contact angle of the elements with each other affected the work of the entire contact system. This started from the forces required to close the system, through the vibration speeds of the lamellas after closing the contacts, to damage and contact destruction by fault usage. the damage described showed examples of faults that can occur in tulip contacts.

The field distribution simulation studies show the symmetry of the charge distribution with respect to the axis of the tulip contact. Due to the observation of such regularity, several approximations were made in a further analysis. the described cases should be treated as homogeneous movement of the apparatus. Performing such an observation allows the influence of the electric field on individual elements to be observed.

## 10. Conclusions

The procured simulations have shown that the tulip contact is a complex structure and its design for given electrical parameters is the sum of compromises between the individual component values of the entire operating system. the combination of two types of analyzes—motion analysis and analysis of the electric field distribution—gives the image most similar to the real operating environment of the system. This allows the entire contact to be designed in the most optimal way. Optimization concerns both the design work time and the cost and quantity of prototyping. the advantage of simulation analysis is not only the speed of the obtained result, but also the possibility of a very quick modification, either in the model or the set parameters. the obtained results confirm the operation and maintenance documents (DTR) regarding the proper servicing and diagnosis of switching devices. the issue of analyzing the operation of a tulip contact has certainly not been exhausted.

The tulip contact is a very well-designed contact system for switching high-intensity currents and high voltages. Due to the large connection area resulting from many individual elements, the system allows the application of high closing forces, which translates into velocity of the operation. a very good electric field distribution promotes the dielectric properties of the system and improves its electrical parameters. Undoubtedly, numerical analyzes help to select materials and parameters for the system operation very quickly. the tulip contact is a construction that is still used in the professional power industry. It is modified according to current needs. These elements work in an environment of chambers filled with both air and technical gases, such as $SF_6$. the gases allow the operating voltage of the system and the rated and fault currents to be increased thanks to the higher heat dissipation coefficient.

Unfortunately, the current computing environments do not allow analyzes to be carried out as one analysis—analysis of the movement of connections with the applied potentials and material permeabilities. This would reproduce the working environment of the contacts. the official response from ANSYS says that it is currently not possible to combine Explicit Dynamics' high-speed analysis module with ANSYS MAXWELL's field analysis. the procured model of the tulip contact system is complex and as the validation results showed, it can be used for a cost effective method of testing and designing such systems.

Figure 34 clearly showed the significant convergence of results derived from simulations and those captured during laboratory tests. Therefore, the design modifications and further work projects can already be implemented using the proposed simulation. It has been demonstrated that the derived simulation is useful for the rapid changing of modules used in the analysis of the dynamics of tulip contact elements' movement.

**Author Contributions:** Conceptualization, Ł.K. (Łukasz Kolimas), S.Ł., M.S., Ł.K. (Łukasz Kozarek), K.G.; Data curation, Ł.K. (Łukasz Kolimas), S.Ł., M.S., Ł.K. (Łukasz Kozarek); Formal analysis, Ł.K. (Łukasz Kolimas), S.Ł., M.S., Ł.K. (Łukasz Kozarek), K.G.; Investigation, Ł.K. (Łukasz Kolimas), S.Ł., M.S., Ł.K. (Łukasz Kozarek), K.G.; Methodology, Ł.K. (Łukasz Kolimas), S.Ł., M.S., Ł.K. (Łukasz Kozarek), K.G.; Project administration, Ł.K. (Łukasz Kolimas), S.Ł., M.S., Ł.K. (Łukasz Kozarek), K.G.; Resources, Ł.K. (Łukasz Kolimas), S.Ł., M.S., Ł.K. (Łukasz Kozarek), K.G.; Supervision, Ł.K. (Łukasz Kolimas), S.Ł., M.S., Ł.K. (Łukasz Kozarek), K.G.; Validation, Ł.K. (Łukasz Kolimas), S.Ł., M.S., Ł.K. (Łukasz Kozarek), K.G.; Visualization, Ł.K. (Łukasz Kolimas), S.Ł., M.S., Ł.K. (Łukasz Kozarek), K.G.; Writing—original draft, Ł.K. (Łukasz Kolimas), S.Ł., M.S., Ł.K. (Łukasz Kozarek), K.G.;

Writing—review & editing, Ł.K. (Łukasz Kolimas), S.Ł., M.S., Ł.K. (Łukasz Kozarek), K.G.; All authors have read and agreed to the published version of the manuscript.

**Funding:** This research received no external funding.

**Conflicts of Interest:** The authors declare no conflict of interest.

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
