# Peer review of "Mechanical and Electrical Simulations of the Tulip Contact System"

_energies, doi:10.3390/en13195059_

Round 1

Reviewer 1 Report

The novelty of the paper is not clearly described. Therefore, based on the paper's motivation or contribution, the paper impact cannot be judged high.

I have some concerns to be satisfied before paper publication:

  • The introduction section is very weak and short. The references are cited in the merging format and the authors did not describe the literature review separately.
  • Where are the references [1] and [2] in the start of the introduction? The references number is started with [3]! Please sort the used references by the number.
  • What's the author's main contribution? In the introduction section, the authors should state the weakness of the previous researches and propose their contribution compared to them. It is better that the authors highlight the contributions with a bulleted list.
  • In page 3 and paragraph 1, is really needed to cite 7 references [25-31] for this simple subject? It seems that the authors did not spend enough time to consider and select the best references for every issue.
  • I think it not necessary to show everything by the figure, because some of them are well-known for the readers. For example figure 2 is not mandatory to illustrate. They can be only cited by the authors.
  • Please use the standard writing format for the figures and tables. For example, you wrote " Full material data used in the simulation is described in the table: Table 1 above." In page 4 and "The example of such an employment is shown in Figure 2 below." In page 3. It is better to write only "table 1 describes the full material data in the simulation ".
  • In the page 5 and paragraph 3, are the information derived by the authors or the references? (Information about Sulfur hexafluoride and SF6). If they are extracted by the references, please cite them.
  • It is not needed to allocate a section (Section 5) for the "Environment for simulation research" with only 1 paragraph. This section can be easily merged with other parts.
  • In the abstract, the authors stated "Concluded work is important on the account of the cost effectiveness for design procedures concerning tulip contacts". However, I did not see the cost analysis in the article. Please clarify.
  • The simulation part is too long and can be summarized by the authors. Please avoid describing too much and try to show your outstanding results.
  • In theory analysis, the relation between the body of the paper and the results is not described. Please connect your method to the simulation test.

Author Response

Dear reviewer,
Thank you very much for the positive and comprehensive comments that helped a lot to improve this paper. All comments were answered and issues corrected or explained.

Reviewer 2 Report

The paper presents simulations of a special type of electrical contact called a tulip contact. Thus, the article addresses all strategies that allow simulating the performance of the electrical contact. The paper makes a detailed evaluation of the characteristics of the electrical contact such as the construction details, the physical properties, the environment used for simulations and the results obtained in simulations. Thus, the paper is interesting and pleasant to read. Thus, this reviewer has no conceptual observations. However, this reviewer has the following secondary observations: (i) authors should be careful to mention compact notations the first time they are used (see, for example, FEM (Finite Element Method); (ii) a general review is recommended of the numbering of the figures and the calls of the figures (see that figures 14, 15 and 16 are called before Figure 13 and in line 505 the call of figures 28, 29 and 30 are wrong, that is, figures 29, 30 and 31 must be called), (iii) the reference [3] appears to be incomplete, and (iv) a general revision of the document to eliminate trivial editing errors.

Author Response

Dear reviewer,
Thank you very much for positive and comprehensive comments that helped a lot to improve this paper. All comments were answered and issues corrected or explained.

Round 2

Reviewer 1 Report

I have no other technical comments. Thank you for your corrections.

Accept.